# Matryoshka Representation Learning

**Aditya Kusupati**[*†◇]**, Gantavya Bhatt**[*†]**, Aniket Rege**[*†]**,**
**Matthew Wallingford**[†]**, Aditya Sinha**[◇]**, Vivek Ramanujan**[†]**, William Howard-Snyder**[†]**,**
**Kaifeng Chen**[◇]**, Sham Kakade**[‡]**, Prateek Jain**[◇] **and Ali Farhadi**[†]
[†]University of Washington, [◇]Google Research, [‡]Harvard University
{kusupati,ali}@cs.washington.edu, prajain@google.com

## Abstract

Learned representations are a central component in modern ML systems, serving a multitude of downstream tasks. When training such representations, it is often the case that computational and statistical constraints for each downstream task are unknown. In this context, rigid fixed-capacity representations can be either over or under-accommodating to the task at hand. This leads us to ask: *can we design a flexible representation that can adapt to multiple downstream tasks with varying computational resources?* Our main contribution is 🪆 Matryoshka Representation Learning (MRL) which encodes information at different granularities and allows a single embedding to adapt to the computational constraints of downstream tasks. MRL minimally modifies existing representation learning pipelines and imposes no additional cost during inference and deployment. MRL learns coarse-to-fine representations that are at least as accurate and rich as independently trained low-dimensional representations. The flexibility within the learned Matryoshka Representations offer: (a) up to $14\times$ smaller embedding size for ImageNet-1K classification at the same level of accuracy; (b) up to $14\times$ real-world speed-ups for large-scale retrieval on ImageNet-1K and 4K; and (c) up to $2\%$ accuracy improvements for long-tail few-shot classification, all while being as robust as the original representations. Finally, we show that MRL extends seamlessly to web-scale datasets (ImageNet, JFT) across various modalities – vision (ViT, ResNet), vision + language (ALIGN) and language (BERT). MRL code and pretrained models are open-sourced at https://github.com/RAIVNLab/MRL.

## 1 Introduction

Learned representations [55] are fundamental building blocks of real-world ML systems [62, 86]. Trained once and frozen, $d$-dimensional representations encode rich information and can be used to perform multiple downstream tasks [4]. The deployment of deep representations has two steps: (1) an expensive yet constant-cost forward pass to compute the representation [27] and (2) utilization of the representation for downstream applications [48, 84]. Compute costs for the latter part of the pipeline scale with the embedding dimensionality as well as the data size ($N$) and label space ($L$). At web-scale [15, 80] this utilization cost overshadows the feature computation cost. The rigidity in these representations forces the use of high-dimensional embedding vectors across multiple tasks despite the varying resource and accuracy constraints that require flexibility.

Human perception of the natural world has a naturally coarse-to-fine granularity [26, 30]. However, perhaps due to the inductive bias of gradient-based training [79], deep learning models tend to diffuse "information" across the entire representation vector. The desired elasticity is usually enabled in the existing flat and fixed representations either through training multiple low-dimensional models [27], jointly optimizing sub-networks of varying capacity [9, 95] or post-hoc compression [36, 57]. Each of these techniques struggle to meet the requirements for adaptive large-scale deployment either

---

[*]Equal contribution – AK led the project with extensive support from GB and AR for experimentation.

36th Conference on Neural Information Processing Systems (NeurIPS 2022).

due to training/maintenance overhead, numerous expensive forward passes through all of the data, storage and memory cost for multiple copies of encoded data, expensive on-the-fly feature selection or a significant drop in accuracy. By encoding coarse-to-fine-grained representations, which are as accurate as the independently trained counterparts, we learn with minimal overhead a representation that can be deployed *adaptively* at no additional cost during inference.

We introduce 🪆 Matryoshka Representation Learning (MRL) to induce flexibility in the learned representation. MRL learns representations of varying capacities within the same high-dimensional vector through explicit optimization of $O(\log(d))$ lower-dimensional vectors in a nested fashion, hence the name Matryoshka. MRL can be adapted to any existing representation pipeline and is easily extended to many standard tasks in computer vision and natural language processing. Figure 1 illustrates the core idea of Matryoshka Representation Learning (MRL) and the adaptive deployment settings of the learned Matryoshka Representations.

The first $m$-dimensions, $m \in [d]$, of the Matryoshka Representation is an information-rich low-dimensional vector, at no additional training cost, that is as accurate as an independently trained $m$-dimensional representation. The information within the Matryoshka Representation increases with the dimensionality creating a coarse-to-fine grained representation, all without significant training or additional deployment overhead. MRL equips the representation vector with the desired flexibility and multi-fidelity that can ensure a near-optimal accuracy-vs-compute trade-off. With these advantages, MRL enables adaptive deployment based on accuracy and compute constraints.

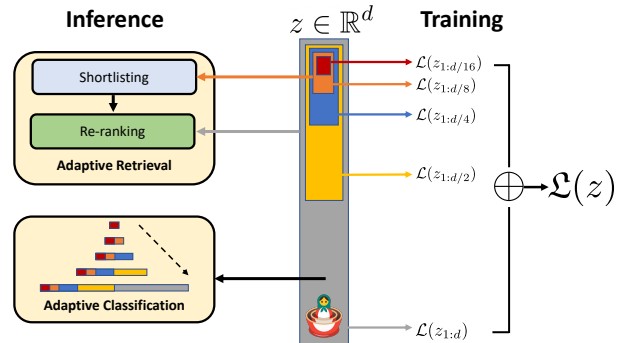

Figure 1: 🪆 Matryoshka Representation Learning is adaptable to any representation learning setup and begets a Matryoshka Representation $z$ by optimizing the original loss $\mathcal{L}(.)$ at $O(\log(d))$ chosen representation sizes. Matryoshka Representation can be utilized effectively for adaptive deployment across environments and downstream tasks.

The Matryoshka Representations improve efficiency for large-scale classification and retrieval without any significant loss of accuracy. While there are potentially several applications of coarse-to-fine Matryoshka Representations, in this work we focus on two key building blocks of real-world ML systems: large-scale classification and retrieval. For classification, we use adaptive cascades with the variable-size representations from a model trained with MRL, significantly reducing the average dimension of embeddings needed to achieve a particular accuracy. For example, on ImageNet-1K, MRL + adaptive classification results in up to a $14\times$ smaller representation size at the same accuracy as baselines (Section 4.2.1). Similarly, we use MRL in an adaptive retrieval system. Given a query, we shortlist retrieval candidates using the first few dimensions of the query embedding, and then successively use more dimensions to re-rank the retrieved set. A simple implementation of this approach leads to $128\times$ theoretical (in terms of FLOPS) and $14\times$ wall-clock time speedups compared to a single-shot retrieval system that uses a standard embedding vector; note that MRL's retrieval accuracy is comparable to that of single-shot retrieval (Section 4.3.1). Finally, as MRL explicitly learns coarse-to-fine representation vectors, intuitively it should share more semantic information among its various dimensions (Figure 5). This is reflected in up to $2\%$ accuracy gains in long-tail continual learning settings while being as robust as the original embeddings. Furthermore, due to its coarse-to-fine grained nature, MRL can also be used as method to analyze hardness of classification among instances and information bottlenecks.

**We make the following key contributions:**

1. We introduce 🪆 Matryoshka Representation Learning (MRL) to obtain flexible representations (Matryoshka Representations) for adaptive deployment (Section 3).
2. Up to $14\times$ faster yet accurate large-scale classification and retrieval using MRL (Section 4).
3. Seamless adaptation of MRL across modalities (vision - ResNet & ViT, vision + language - ALIGN, language - BERT) and to web-scale data (ImageNet-1K/4K, JFT-300M and ALIGN data).
4. Further analysis of MRL's representations in the context of other downstream tasks (Section 5).

## 2   Related Work

**Representation Learning.**   Large-scale datasets like ImageNet [16, 71] and JFT [80] enabled the learning of general purpose representations for computer vision [4, 93]. These representations are typically learned through supervised and un/self-supervised learning paradigms. Supervised pretraining [27, 49, 77] casts representation learning as a multi-class/label classification problem, while un/self-supervised learning learns representation via proxy tasks like instance classification [92] and reconstruction [29, 60]. Recent advances [12, 28] in contrastive learning [25] enabled learning from web-scale data [21] that powers large-capacity cross-modal models [18, 44, 67, 96]. Similarly, natural language applications are built [38] on large language models [8] that are pretrained [64, 70] in a un/self-supervised fashion with masked language modelling [19] or autoregressive training [66].

🪆 Matryoshka Representation Learning (MRL) is complementary to all these setups and can be adapted with minimal overhead (Section 3). MRL equips representations with multifidelity at no additional cost which enables adaptive deployment based on the data and task (Section 4).

**Efficient Classification and Retrieval.**   Efficiency in classification and retrieval during inference can be studied with respect to the high yet constant deep featurization costs or the search cost which scales with the size of the label space and data. Efficient neural networks address the first issue through a variety of algorithms [24, 52] and design choices [37, 51, 82]. However, with a strong featurizer, most of the issues with scale are due to the linear dependence on number of labels ($L$), size of the data ($N$) and representation size ($d$), stressing RAM, disk and processor all at the same time.

The sub-linear complexity dependence on number of labels has been well studied in context of compute [3, 41, 65] and memory [20] using Approximate Nearest Neighbor Search (ANNS) [59] or leveraging the underlying hierarchy [17, 53]. In case of the representation size, often dimensionality reduction [72, 83], hashing techniques [14, 50, 73] and feature selection [61] help in alleviating selective aspects of the $O(d)$ scaling at a cost of significant drops in accuracy. Lastly, most real-world search systems [11, 15] are often powered by large-scale embedding based retrieval [10, 62] that scales in cost with the ever increasing web-data. While categorization [84, 94] clusters similar things together, it is imperative to be equipped with retrieval capabilities that can bring forward every instance [7]. Approximate Nearest Neighbor Search (ANNS) [40] makes it feasible with efficient indexing [14] and traversal [5, 6] to present the users with the most similar documents/images from the database for a requested query. Widely adopted HNSW [59] ($O(d \log(N))$) is as accurate as exact retrieval ($O(dN)$) at the cost of a graph-based index overhead for RAM and disk [42].

MRL tackles the linear dependence on embedding size, $d$, by learning multifidelity Matryoshka Representations. Lower-dimensional Matryoshka Representations are as accurate as independently trained counterparts without the multiple expensive forward passes. Matryoshka Representations provide an *intermediate abstraction* between high-dimensional vectors and their efficient ANNS indices through the adaptive embeddings nested within the original representation vector (Section 4). All other aforementioned efficiency techniques are complementary and can be readily applied to the learned Matryoshka Representations obtained from MRL.

Several works in efficient neural network literature [9, 88, 95] aim at packing neural networks of varying capacity within the same larger network. However, the weights for each progressively smaller network can be different and often require distinct forward passes to isolate the final representations. This is detrimental for adaptive inference due to the need for re-encoding the entire retrieval database with expensive sub-net forward passes of varying capacities. Finally, ordered representations proposed by Rippel et al. [69] use nested dropout in the context of autoencoders to learn nested representations. MRL differentiates itself in formulation by optimizing only for $O(\log(d))$ nesting dimensions instead of $O(d)$. Despite this, MRL diffuses information to intermediate dimensions interpolating between the optimized Matryoshka Representation sizes accurately (Figure 5); making web-scale feasible.

## 3   🪆 Matryoshka Representation Learning

For $d \in \mathbb{N}$, consider a set $\mathcal{M} \subset [d]$ of representation sizes. For a datapoint $x$ in the input domain $\mathcal{X}$, our goal is to learn a $d$-dimensional representation vector $z \in \mathbb{R}^d$. For every $m \in \mathcal{M}$, Matryoshka Representation Learning (MRL) enables each of the first $m$ dimensions of the embedding vector, $z_{1:m} \in \mathbb{R}^m$ to be independently capable of being a transferable and general purpose

representation of the datapoint $x$. We obtain $z$ using a deep neural network $F(\,\cdot\,;\theta_F)\colon \mathcal{X} \to \mathbb{R}^d$ parameterized by learnable weights $\theta_F$, i.e., $z \coloneqq F(x;\theta_F)$. The multi-granularity is captured through the set of the chosen dimensions $\mathcal{M}$, that contains less than $\log(d)$ elements, i.e., $|\mathcal{M}| \leq \lfloor \log(d) \rfloor$. The usual set $\mathcal{M}$ consists of consistent halving until the representation size hits a low information bottleneck. We discuss the design choices in Section 4 for each of the representation learning settings.

For the ease of exposition, we present the formulation for fully supervised representation learning via multi-class classification. Matryoshka Representation Learning modifies the typical setting to become a multi-scale representation learning problem on the same task. For example, we train ResNet50 [27] on ImageNet-1K [71] which embeds a $224 \times 224$ pixel image into a $d = 2048$ representation vector and then passed through a linear classifier to make a prediction, $\hat{y}$ among the $L = 1000$ labels. For MRL, we choose $\mathcal{M} = \{8, 16, \ldots, 1024, 2048\}$ as the nesting dimensions.

Suppose we are given a labelled dataset $\mathcal{D} = \{(x_1, y_1), \ldots, (x_N, y_N)\}$ where $x_i \in \mathcal{X}$ is an input point and $y_i \in [L]$ is the label of $x_i$ for all $i \in [N]$. MRL optimizes the multi-class classification loss for each of the nested dimension $m \in \mathcal{M}$ using standard empirical risk minimization using a separate linear classifier, parameterized by $\mathbf{W}^{(m)} \in \mathbb{R}^{L \times m}$. All the losses are aggregated after scaling with their relative importance $(c_m \geq 0)_{m \in \mathcal{M}}$ respectively. That is, we solve

$$\min_{\{\mathbf{W}^{(m)}\}_{m \in \mathcal{M}},\ \theta_F} \frac{1}{N} \sum_{i \in [N]} \sum_{m \in \mathcal{M}} c_m \cdot \mathcal{L}\left(\mathbf{W}^{(m)} \cdot F(x_i;\theta_F)_{1:m}\ ;\ y_i\right)\ , \qquad (1)$$

where $\mathcal{L}\colon \mathbb{R}^L \times [L] \to \mathbb{R}_+$ is the multi-class softmax cross-entropy loss function. This is a standard optimization problem that can be solved using sub-gradient descent methods. We set all the importance scales, $c_m = 1$ for all $m \in \mathcal{M}$; see Section 5 for ablations. Lastly, despite only optimizing for $O(\log(d))$ nested dimensions, MRL results in accurate representations, that interpolate, for dimensions that fall between the chosen granularity of the representations (Section 4.2).

We call this formulation as Matryoshka Representation Learning (MRL). A natural way to make this efficient is through weight-tying across all the linear classifiers, i.e., by defining $\mathbf{W}^{(m)} = \mathbf{W}_{1:m}$ for a set of common weights $\mathbf{W} \in \mathbb{R}^{L \times d}$. This would reduce the memory cost due to the linear classifiers by almost half, which would be crucial in cases of extremely large output spaces [84, 94]. This variant is called *Efficient* Matryoshka Representation Learning (MRL–E). Refer to Alg 1 and Alg 2 in Appendix A for the building blocks of Matryoshka Representation Learning (MRL).

**Adaptation to Learning Frameworks.** MRL can be adapted seamlessly to most representation learning frameworks at web-scale with minimal modifications (Section 4.1). For example, MRL's adaptation to masked language modelling reduces to MRL–E due to the weight-tying between the input embedding matrix and the linear classifier. For contrastive learning, both in context of vision & vision + language, MRL is applied to both the embeddings that are being contrasted with each other. The presence of normalization on the representation needs to be handled independently for each of the nesting dimension for best results (see Appendix C for more details).

## 4 Applications

In this section, we discuss Matryoshka Representation Learning (MRL) for a diverse set of applications along with an extensive evaluation of the learned multifidelity representations. Further, we showcase the downstream applications of the learned Matryoshka Representations for flexible large-scale deployment through (a) Adaptive Classification (AC) and (b) Adaptive Retrieval (AR).

### 4.1 Representation Learning

We adapt Matryoshka Representation Learning (MRL) to various representation learning setups (a) Supervised learning for vision: ResNet50 [27] on ImageNet-1K [71] and ViT-B/16 [22] on JFT-300M [80], (b) Contrastive learning for vision + language: ALIGN model with ViT-B/16 vision encoder and BERT language encoder on ALIGN data [44] and (c) Masked language modelling: BERT [19] on English Wikipedia and BooksCorpus [97]. Please refer to Appendices B and C for details regarding the model architectures, datasets and training specifics.

We do not search for best hyper-parameters for all MRL experiments but use the same hyper-parameters as the independently trained baselines. ResNet50 outputs a 2048-dimensional repre-

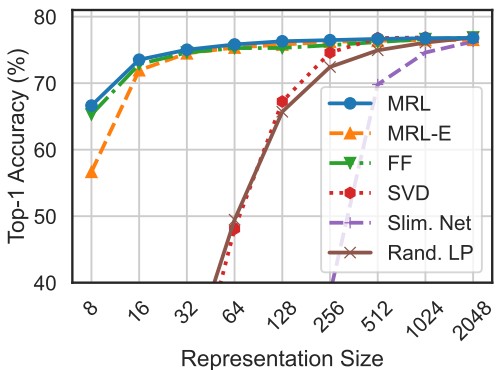
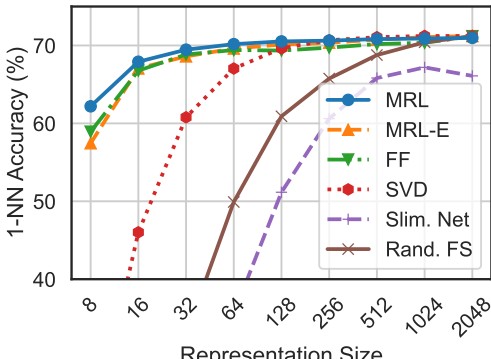

Figure 2: ImageNet-1K linear classification accuracy of ResNet50 models. MRL is as accurate as the independently trained FF models for every representation size.

Figure 3: ImageNet-1K 1-NN accuracy of ResNet50 models measuring the representation quality for downstream task. MRL outperforms all the baselines across all representation sizes.

sentation while ViT-B/16 and BERT-Base output 768-dimensional embeddings for each data point. We use $\mathcal{M} = \{8, 16, 32, 64, 128, 256, 512, 1024, 2048\}$ and $\mathcal{M} = \{12, 24, 48, 96, 192, 384, 768\}$ as the explicitly optimized nested dimensions respectively. Lastly, we extensively compare the MRL and MRL–E models to independently trained low-dimensional (fixed feature) representations (FF), dimensionality reduction (SVD), sub-net method (slimmable networks [95]) and randomly selected features of the highest capacity FF model.

In section 4.2, we evaluate the quality and capacity of the learned representations through linear classification/probe (LP) and 1-nearest neighbour (1-NN) accuracy. Experiments show that MRL models remove the dependence on $|\mathcal{M}|$ resource-intensive independently trained models for the coarse-to-fine representations while being as accurate. Lastly, we show that despite optimizing only for $|\mathcal{M}|$ dimensions, MRL models diffuse the information, in an interpolative fashion, across all the $d$ dimensions providing the finest granularity required for adaptive deployment.

## 4.2 Classification

Figure 2 compares the linear classification accuracy of ResNet50 models trained and evaluated on ImageNet-1K. ResNet50–MRL model is at least as accurate as each FF model at every representation size in $\mathcal{M}$ while MRL–E is within $1\%$ starting from 16-dim. Similarly, Figure 3 showcases the comparison of learned representation quality through 1-NN accuracy on ImageNet-1K (trainset with 1.3M samples as the database and validation set with 50K samples as the queries). Matryoshka Representations are up to $2\%$ more accurate than their fixed-feature counterparts for the lower-dimensions while being as accurate elsewhere. 1-NN accuracy is an excellent proxy, at no additional training cost, to gauge the utility of learned representations in the downstream tasks.

We also evaluate the quality of the representations from training ViT-B/16 on JFT-300M alongside the ViT-B/16 vision encoder of the ALIGN model – two web-scale setups. Due to the expensive nature of these experiments, we only train the highest capacity fixed feature model and choose random features for evaluation in lower-dimensions. Web-scale is a compelling setting for MRL due to its relatively inexpensive training overhead while providing multifidelity representations for downstream tasks. Figure 4, evaluated with 1-NN on ImageNet-1K, shows that all the MRL models for JFT and ALIGN are highly accurate while providing an excellent cost-vs-accuracy trade-off at lower-dimensions. These experiments show that MRL seamlessly scales to large-scale models and web-scale datasets while providing the otherwise prohibitively expensive multi-granularity in the process. We also have similar observations when pretraining BERT; please see Appendix D.2 for more details. Our experiments also show that post-hoc compression (SVD), linear probe on random features, and sub-net style slimmable networks drastically lose accuracy compared to MRL as the representation size decreases. Finally, Figure 5 shows that, while MRL explicitly optimizes $O(\log(d))$ nested representations – removing the $O(d)$ dependence [69] –, the coarse-to-fine grained information is interpolated across all $d$ dimensions providing highest flexibility for adaptive deployment.

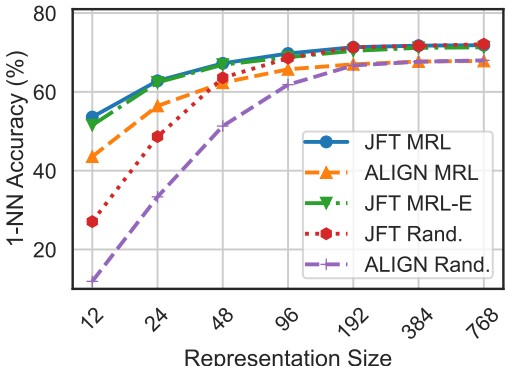
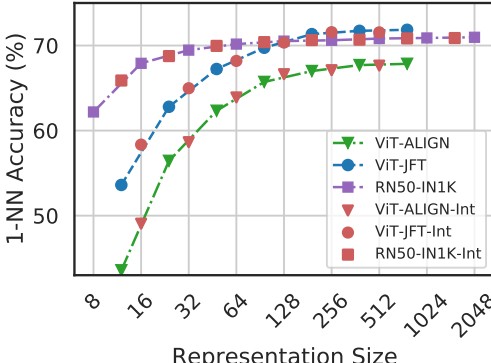

Figure 4: ImageNet-1K 1-NN accuracy for ViT-B/16 models trained on JFT-300M & as part of ALIGN. MRL scales seamlessly to web-scale with minimal training overhead.

Figure 5: Despite optimizing MRL only for $O(\log(d))$ dimensions for ResNet50 and ViT-B/16 models; the accuracy in the intermediate dimensions shows interpolating behaviour.

#### 4.2.1 Adaptive Classification

The flexibility and coarse-to-fine granularity within Matryoshka Representations allows model cascades [85] for Adaptive Classification (AC) [26]. Unlike standard model cascades [90], MRL does not require multiple expensive neural network forward passes. To perform AC with an MRL trained model, we learn thresholds on the maximum softmax probability [31] for each nested classifier on a holdout validation set. We then use these thresholds to decide when to transition to the higher dimensional representation (e.g $8 \rightarrow 16 \rightarrow 32$) of the MRL model. Appendix D.1 discusses the implementation and learning of thresholds for cascades used for adaptive classification in detail.

Figure 6 shows the comparison between cascaded MRL representations (MRL–AC) and independently trained fixed feature (FF) models on ImageNet-1K with ResNet50. We computed the expected representation size for MRL–AC based on the final dimensionality used in the cascade. We observed that MRL–AC was as accurate, 76.30%, as a 512-dimensional FF model but required an expected dimensionality of $\sim 37$ while being only $0.8\%$ lower than the 2048-dimensional FF baseline. Note that all MRL–AC models are significantly more accurate than the FF baselines at comparable representation sizes. MRL–AC uses up to $\sim 14\times$ smaller representation size for the same accuracy which affords computational efficiency as the label space grows [84]. Lastly, our results with MRL–AC indicate that instances and classes vary in difficulty which we analyze in Section 5 and Appendix J.

### 4.3 Retrieval

Nearest neighbour search with learned representations powers a plethora of retrieval and search applications [15, 86, 11, 62]. In this section, we discuss the image retrieval performance of the pretrained ResNet50 models (Section 4.1) on two large-scale datasets ImageNet-1K [71] and ImageNet-4K. ImageNet-1K has a database size of ~1.3M and a query set of 50K samples uniformly spanning 1000 classes. We also introduce ImageNet-4K which has a database size of ~4.2M and query set of ~200K samples uniformly spanning 4202 classes (see Appendix B for details). A single forward pass on ResNet50 costs 4 GFLOPs while exact retrieval costs 2.6 GFLOPs per query for ImageNet-1K. Although retrieval overhead is $40\%$ of the total cost, retrieval cost grows linearly with the size of the database. ImageNet-4K presents a retrieval benchmark where the exact search cost becomes the computational bottleneck (8.6 GFLOPs per query). In both these settings, the memory and disk usage are also often bottlenecked by the large databases. However, in most real-world applications exact search, $O(dN)$, is replaced with an approximate nearest neighbor search (ANNS) method like HNSW [59], $O(d\log(N))$, with minimal accuracy drop at the cost of additional memory overhead.

The goal of image retrieval is to find images that belong to the same class as the query using representations obtained from a pretrained model. In this section, we compare retrieval performance using mean Average Precision @ 10 (mAP@10) which comprehensively captures the setup of relevant image retrieval at scale. We measure the cost per query using exact search in MFLOPs. All embeddings are unit normalized and retrieved using the L2 distance metric. Lastly, we report

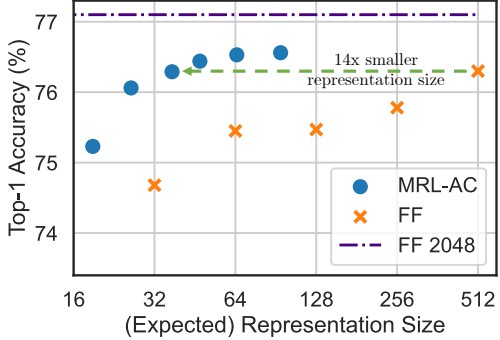
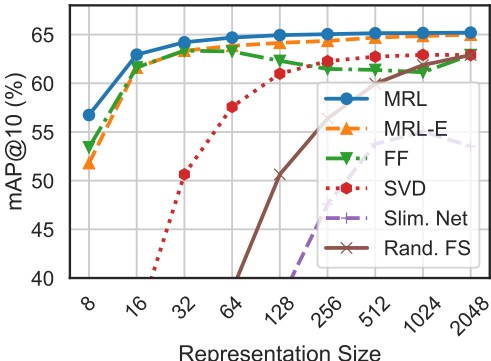

Figure 6: Adaptive classification on MRL ResNet50 using cascades results in $14\times$ smaller representation size for the same level of accuracy on ImageNet-1K ($\sim 37$ vs $512$ dims for $76.3\%$).

Figure 7: mAP@10 for Image Retrieval on ImageNet-1K with ResNet50. MRL consistently produces better retrieval performance over the baselines across all the representation sizes.

an extensive set of metrics spanning mAP@$k$ and P@$k$ for $k = \{10, 25, 50, 100\}$ and real-world wall-clock times for exact search and HNSW. See Appendices E and F for more details.

Figure 7 compares the mAP@10 performance of ResNet50 representations on ImageNet-1K across dimensionalities for MRL, MRL–E, FF, slimmable networks along with post-hoc compression of vectors using SVD and random feature selection. Matryoshka Representations are often the most accurate while being up to $3\%$ better than the FF baselines. Similar to classification, post-hoc compression and slimmable network baselines suffer from significant drop-off in retrieval mAP@10 with $\leq 256$ dimensions. Appendix E discusses the mAP@10 of the same models on ImageNet-4K.

MRL models are capable of performing accurate retrieval at various granularities without the additional expense of multiple model forward passes for the web-scale databases. FF models also generate independent databases which become prohibitively expense to store and switch in between. Matryoshka Representations enable adaptive retrieval (AR) which alleviates the need to use full-capacity representations, $d = 2048$, for all data and downstream tasks. Lastly, all the vector compression techniques [57, 43] used as part of the ANNS pipelines are complimentary to Matryoshka Representations and can further improve the efficiency-vs-accuracy trade-off.

### 4.3.1 Adaptive Retrieval

We benchmark MRL in the adaptive retrieval setting (AR) [48]. For a given query image, we obtained a shortlist, $K = 200$, of images from the database using a lower-dimensional representation, e.g. $D_s = 16$ followed by reranking with a higher capacity representation, e.g. $D_r = 2048$. In real-world scenarios where top ranking performance is the key objective, measured with mAP@$k$ where k covers a limited yet crucial real-estate, AR provides significant compute and memory gains over single-shot retrieval with representations of fixed dimensionality. Finally, the most expensive part of AR, as with any retrieval pipeline, is the nearest neighbour search for shortlisting. For example, even naive re-ranking of 200 images with 2048 dimensions only costs 400 KFLOPs. While we report exact search cost per query for all AR experiments, the shortlisting component of the pipeline can be sped-up using ANNS (HNSW). Appendix I has a detailed discussion on compute cost for exact search, memory overhead of HNSW indices and wall-clock times for both implementations. We note that using HNSW with 32 neighbours for shortlisting does not decrease accuracy during retrieval.

Figure 8 showcases the compute-vs-accuracy trade-off for adaptive retrieval using Matryoshka Representations compared to single-shot using fixed features with ResNet50 on ImageNet-1K. We observed that all AR settings lied above the Pareto frontier of single-shot retrieval with varying representation sizes. In particular for ImageNet-1K, we show that the AR model with $D_s = 16$ & $D_r = 2048$ is as accurate as single-shot retrieval with $d = 2048$ while being $\sim \mathbf{128\times}$ more efficient in theory and $\sim \mathbf{14\times}$ faster in practice (compared using HNSW on the same hardware). We show similar trends with ImageNet-4K, but note that we require $D_s = 64$ given the increased difficulty of the dataset. This results in $\sim 32\times$ and $\sim 6\times$ theoretical and in-practice speedups respectively. Lastly, while $K = 200$ works well for our adaptive retrieval experiments, we ablated over the shortlist size $k$ in Appendix K.2 and found that the accuracy gains stopped after a

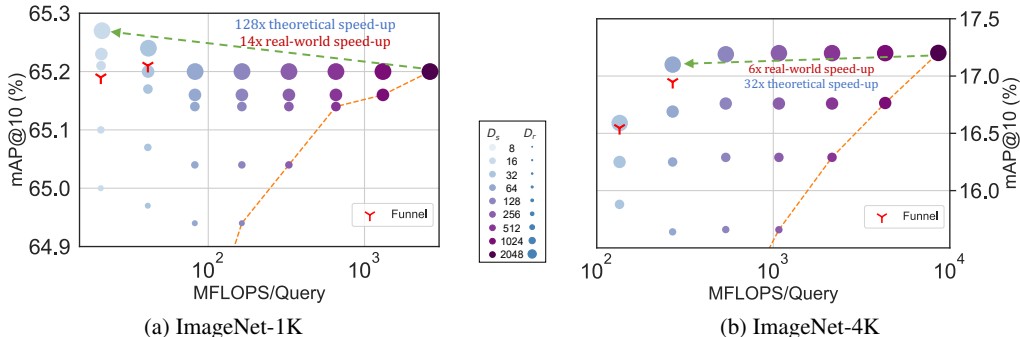

(a) ImageNet-1K        (b) ImageNet-4K

Figure 8: The trade-off between mAP@$10$ vs MFLOPs/Query for Adaptive Retrieval (AR) on ImageNet-1K (left) and ImageNet-4K (right). Every combination of $D_s$ & $D_r$ falls above the Pareto line (orange dots) of single-shot retrieval with a fixed representation size while having configurations that are as accurate while being up to $14\times$ faster in real-world deployment. Funnel retrieval is almost as accurate as the baseline while alleviating some of the parameter choices of Adaptive Retrieval.

point, further strengthening the use-case for $\mathrm{Matryoshka\ Representation\ Learning}$ and adaptive retrieval.

Even with adaptive retrieval, it is hard to determine the choice of $D_s$ & $D_r$. In order to alleviate this issue to an extent, we propose **Funnel Retrieval**, a consistent cascade for adaptive retrieval. Funnel thins out the initial shortlist by a repeated re-ranking and shortlisting with a series of increasing capacity representations. Funnel halves the shortlist size and doubles the representation size at every step of re-ranking. For example on ImageNet-1K, a funnel with the shortlist progression of $200 \rightarrow 100 \rightarrow 50 \rightarrow 25 \rightarrow 10$ with the cascade of $16 \rightarrow 32 \rightarrow 64 \rightarrow 128 \rightarrow 256 \rightarrow 2048$ representation sizes within $\mathrm{Matryoshka\ Representation}$ is as accurate as the single-shot 2048-dim retrieval while being $\sim 128\times$ more efficient theoretically (see Appendix F for more results). All these results showcase the potential of $\mathrm{MRL}$ and AR for large-scale multi-stage search systems [15].

## 5    Further Analysis and Ablations

**Robustness.**    We evaluate the robustness of the $\mathrm{MRL}$ models trained on ImageNet-1K on out-of-domain datasets, ImageNetV2/R/A/Sketch [68, 32, 33, 89], and compare them to the FF baselines. Table 17 in Appendix H demonstrates that $\mathrm{Matryoshka\ Representations}$ for classification are at least as robust as the original representation while improving the performance on ImageNet-A by $0.6\% - a\ 20\%$ relative improvement. We also study the robustness in the context of retrieval by using ImageNetV2 as the query set for ImageNet-1K database. Table 9 in Appendix E shows that $\mathrm{MRL}$ models have more robust retrieval compared to the FF baselines by having up to $3\%$ higher mAP@$10$ performance. This observation also suggests the need for further investigation into robustness using nearest neighbour based classification and retrieval instead of the standard linear probing setup. We also find that the zero-shot robustness of ALIGN-MRL (Table 18 in Appendix H) agrees with the observations made by Wortsman et al. [91]. Lastly, Table 6 in Appendix D.2 shows that $\mathrm{MRL}$ also improves the cosine similarity span between positive and random image-text pairs.

**Few-shot and Long-tail Learning.**    We exhaustively evaluated few-shot learning on $\mathrm{MRL}$ models using nearest class mean [74]. Table 15 in Appendix G shows that that representations learned through $\mathrm{MRL}$ perform comparably to FF representations across varying shots and number of classes.

$\mathrm{Matryoshka\ Representations}$ realize a unique pattern while evaluating on FLUID [87], a long-tail sequential learning framework. We observed that $\mathrm{MRL}$ provides up to $2\%$ accuracy higher on novel classes in the tail of the distribution, without sacrificing accuracy on other classes (Table 16 in Appendix G). Additionally we find the accuracy between low-dimensional and high-dimensional representations is marginal for pretrain classes. We hypothesize that the higher-dimensional representations are required to differentiate the classes when few training examples of each are known. This results provides further evidence that different tasks require varying capacity based on their difficulty.

**Disagreement across Dimensions.**    The information packing in $\mathrm{Matryoshka\ Representations}$ often results in gradual increase of accuracy with increase in capacity. However, we observed that

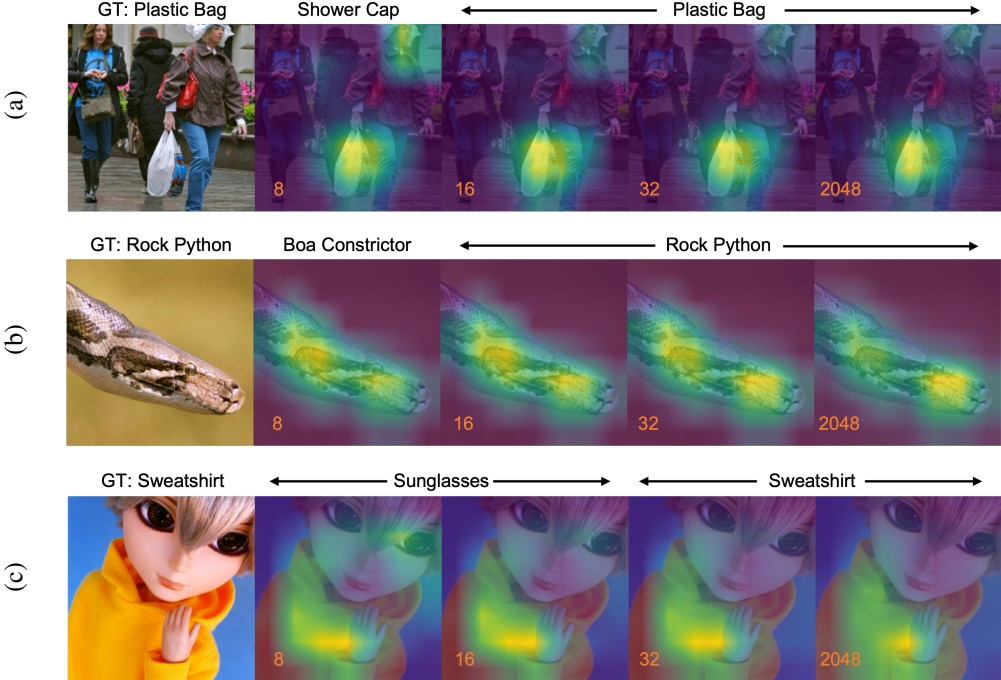

Figure 9: Grad-CAM [75] progression of predictions in MRL model across $8, 16, 32$ and $2048$ dimensions. (a) 8-dimensional representation confuses due to presence of other relevant objects (with a larger field of view) in the scene and predicts "shower cap" ; (b) 8-dim model confuses within the same super-class of "boa" ; (c) 8 and 16-dim models incorrectly focus on the eyes of the doll ("sunglasses") and not the "sweatshirt" which is correctly in focus at higher dimensions; MRL fails gracefully in these scenarios and shows potential use cases of disagreement across dimensions.

this trend was not ubiquitous and certain instances and classes were more accurate when evaluated with lower-dimensions (Figure 12 in Appendix J). With perfect routing of instances to appropriate dimension, MRL can gain up to $4.6\%$ classification accuracy. At the same time, the low-dimensional models are less accurate either due to confusion within the same superclass [23] of the ImageNet hierarchy or presence of multiple objects of interest. Figure 9 showcases 2 such examples for 8-dimensional representation. These results along with Appendix J put forward the potential for MRL to be a systematic framework for analyzing the utility and efficiency of information bottlenecks.

**Superclass Accuracy.** As the information bottleneck becomes smaller, the overall accuracy on fine-grained classes decreases rapidly (Figure 3). However, the drop-off is not as significant when evaluated at a superclass level (Table 24 in Appendix J). Figure 10 presents that this phenomenon

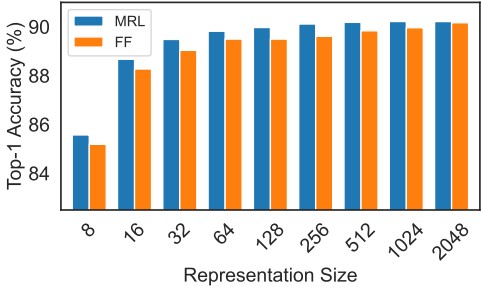

Figure 10: 31-way ImageNet-1K superclass classification across representation size for MRL & FF models showing the capture of underlying hierarchy through tight information bottlenecks.

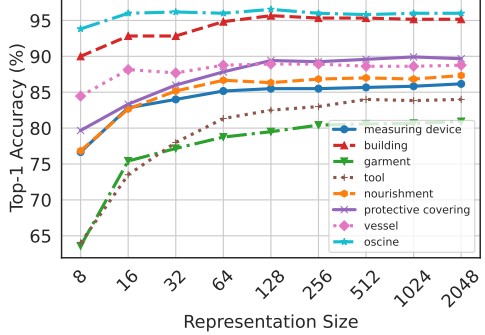

Figure 11: Diverse per-superclass accuracy trends across representation sizes for ResNet50-MRL on ImageNet-1K.

occurs with both MRL and FF models; MRL is more accurate across dimensions. This shows that tight information bottlenecks while not highly accurate for fine-grained classification, do capture required semantic information for coarser classification that could be leveraged for adaptive routing for retrieval and classification. Mutifidelity of Matryoshka Representation naturally captures the underlying hierarchy of the class labels with one single model. Lastly, Figure 11 showcases the accuracy trends per superclass with MRL. The utility of additional dimensions in distinguishing a class from others within the same superclass is evident for "garment" which has up to 11% improvement for $8 \rightarrow 16$ dimensional representation transition. We also observed that superclasses such as "oscine (songbird)" had a clear visual distinction between the object and background and thus predictions using 8 dimensions also led to a good inter-class separability within the superclass.

## 5.1  Ablations

Table 26 in Appendix K presents that Matryoshka Representations can be enabled within off-the-shelf pretrained models with inexpensive partial finetuning thus paving a way for ubiquitous adoption of MRL. At the same time, Table 27 in Appendix C indicates that with optimal weighting of the nested losses we could improve accuracy of lower-dimensions representations without accuracy loss. Tables 28 and 29 in Appendix C ablate over the choice of initial granularity and spacing of the granularites. Table 28 reaffirms the design choice to shun extremely low dimensions that have poor classification accuracy as initial granularity for MRL while Table 29 confirms the effectiveness of logarithmic granularity spacing inspired from the behaviour of accuracy saturation across dimensions over uniform. Lastly, Tables 30 and 31 in Appendix K.2 show that the retrieval performance saturates after a certain shortlist dimension and length depending on the complexity of the dataset.

## 6   Discussion and Conclusions

The results in Section 5.1 reveal interesting weaknesses of MRL that would be logical directions for future work. (1) Optimizing the weightings of the nested losses to obtain a Pareto optimal accuracy-vs-efficiency trade-off – a potential solution could emerge from adaptive loss balancing aspects of anytime neural networks [39]. (2) Using different losses at various fidelities aimed at solving a specific aspect of adaptive deployment – e.g. high recall for 8-dimension and robustness for 2048-dimension. (3) Learning a search data-structure, like differentiable k-d tree, on top of Matryoshka Representation to enable dataset and representation aware retrieval. (4) Finally, the joint optimization of multi-objective MRL combined with end-to-end learnable search data-structure to have data-driven adaptive large-scale retrieval for web-scale search applications.

In conclusion, we presented 🪆 Matryoshka Representation Learning (MRL), a flexible representation learning approach that encodes information at multiple granularities in a single embedding vector. This enables the MRL to adapt to a downstream task's statistical complexity as well as the available compute resources. We demonstrate that MRL can be used for large-scale adaptive classification as well as adaptive retrieval. On standard benchmarks, MRL matches the accuracy of the fixed-feature baseline despite using $14\times$ smaller representation size on average. Furthermore, the Matryoshka Representation based adaptive shortlisting and re-ranking system ensures comparable mAP@10 to the baseline while being $128\times$ cheaper in FLOPs and $14\times$ faster in wall-clock time. Finally, most of the efficiency techniques for model inference and vector search are complementary to MRL 🪆 further assisting in deployment at the compute-extreme environments.

## Acknowledgments

We are grateful to Srinadh Bhojanapalli, Lovish Madaan, Raghav Somani and Ludwig Schmidt for helpful discussions and feedback. Aditya Kusupati also thanks Tom Duerig and Rahul Sukthankar for their support. Part of the paper's large-scale experimentation is supported through a research GCP credit award from Google Cloud and Google Research. Gantavya Bhatt is supported in part by the CONIX Research Center, one of six centers in JUMP, a Semiconductor Research Corporation (SRC) program sponsored by DARPA. Sham Kakade acknowledges funding from the NSF award CCF-1703574 and ONR N00014-22-1-2377. Ali Farhadi acknowledges funding from the NSF awards IIS 1652052, IIS 17303166, DARPA N66001-19-2-4031, DARPA W911NF-15-1-0543 and gifts from Allen Institute for Artificial Intelligence.

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
