# Contents

# A Code for Matryoshka Representation Learning 🪆 (MRL)

We use Alg 1 and 2 provided below to train supervised ResNet50–MRL models on ImageNet-1K. We provide this code as a template to extend MRL to any domain.

---

**Algorithm 1** Pytorch code for Matryoshka Cross-Entropy Loss

---

```python
class Matryoshka_CE_Loss(nn.Module):
    def __init__(self, relative_importance, **kwargs):
        super(Matryoshka_CE_Loss, self).__init__()
        self.criterion = nn.CrossEntropyLoss(**kwargs)
        self.relative_importance = relative_importance # usually set
            to all ones

    def forward(self, output, target):
        loss=0
        for i in range(len(output)):
            loss+= self.relative_importance[i] * self.criterion(output[
                i], target)
        return loss
```

---

---

**Algorithm 2** Pytorch code for MRL Linear Layer

---

```python
class MRL_Linear_Layer(nn.Module):
    def __init__(self, nesting_list: List, num_classes=1000, efficient=
        False, **kwargs):
        super(MRL_Linear_Layer, self).__init__()
        self.nesting_list=nesting_list # set of m in M (Eq. 1)
        self.num_classes=num_classes
        self.is_efficient=efficient # flag for MRL-E

        if not is_efficient:
            for i, num_feat in enumerate(self.nesting_list):
                setattr(self, f"nesting_classifier_{i}", nn.Linear(
                    num_feat, self.num_classes, **kwargs))
        else:
            setattr(self, "nesting_classifier_0", nn.Linear(self.
                nesting_list[-1], self.num_classes, **kwargs)) #
                Instantiating one nn.Linear layer for MRL-E

    def forward(self, x):
        nesting_logits = ()
        for i, num_feat in enumerate(self.nesting_list):
            if(self.is_efficient):
                efficient_logit = torch.matmul(x[:, :num_feat],
                    (self.nesting_classifier_0.weight[:, :
                    num_feat]).t())
            else:
                nesting_logits.append(getattr(self, f"
                    nesting_classifier_{i}")(x[:, :num_feat]))

        if(self.is_efficient):
            nesting_logits.append(efficient_logit)

        return nesting_logits
```

---

# B  Datasets

**ImageNet-1K** [71] contains 1,281,167 labeled train images, and 50,000 labelled validation images across 1,000 classes. The images were transformed with standard procedures detailed by FFCV [54].

**ImageNet-4K** dataset was constructed by selecting 4,202 classes, non-overlapping with ImageNet-1K, from ImageNet-21K [16] with 1,050 or more examples. The train set contains 1,000 examples and the query/validation set contains 50 examples per class totalling to ∼4.2M and ∼200K respectively. We will release the list of images curated together to construct ImageNet-4K.

**JFT-300M** [80] is a large-scale multi-label dataset with 300M images labelled across 18,291 categories.

**ALIGN** [44] utilizes a large scale noisy image-text dataset containing 1.8B image-text pairs.

**ImageNet Robustness Datasets**    We experimented on the following datasets to examine the robustness of MRL models:

**ImageNetV2** [68] is a collection of 10K images sampled a decade after the original construction of ImageNet [16]. ImageNetV2 contains 10 examples each from the 1,000 classes of ImageNet-1K.

**ImageNet-A** [33] contains 7.5K real-world adversarially filtered images from 200 ImageNet-1K classes.

**ImageNet-R** [32] contains 30K artistic image renditions for 200 of the original ImageNet-1K classes.

**ImageNet-Sketch** [89] contains 50K sketches, evenly distributed over all 1,000 ImageNet-1K classes.

**ObjectNet** [2] contains 50K images across 313 object classes, each containing ∼160 images each.

# C  Matryoshka Representation Learning **Model Training**

We trained all ResNet50–MRL models using the efficient dataloaders of FFCV [54]. We utilized the `rn50_40_epochs.yaml` configuration file of FFCV to train all MRL models defined below:

- MRL: ResNet50 model with the fc layer replaced by `MRL_Linear_Layer(efficient=False)`

- MRL–E: ResNet50 model with the fc layer replaced by `MRL_Linear_Layer(efficient=True)`

- FF–k: ResNet50 model with the fc layer replaced by `torch.nn.Linear(k, num_classes)`, where $k \in [8, 16, 32, 64, 128, 256, 512, 1024, 2048]$. We will henceforth refer to these models as simply FF, with the k value denoting representation size.

We trained all ResNet50 models with a learning rate of $0.475$ with a cyclic learning rate schedule [78]. This was after appropriate scaling ($0.25\times$) of the learning rate specified in the configuration file to accommodate for 2xA100 NVIDIA GPUs available for training, compared to the 8xA100 GPUs utilized in the FFCV benchmarks. We trained with a batch size of 256 per GPU, momentum [81] of 0.9, and an SGD optimizer with a weight decay of 1e-4.

Our code (Appendix A) makes minimal modifications to the training pipeline provided by FFCV to learn Matryoshka Representations.

We trained ViT-B/16 models for JFT-300M on a 8x8 cloud TPU pod [47] using Tensorflow [1] with a batchsize of 128 and trained for 300K steps. Similarly, ALIGN models were trained using Tensorflow on 8x8 cloud TPU pod for 1M steps with a batchsize of 64 per TPU. Both these models were trained with adafactor optimizer [76] with a linear learning rate decay starting at 1e-3.

Lastly, we trained a BERT-Base model on English Wikipedia and BookCorpus. We trained our models in Tensorflow using a 4x4 cloud TPU pod with a total batchsize of 1024. We used AdamW [58] optimizer with a linear learning rate decay starting at 1e-4 and trained for 450K steps.

In each configuration/case, if the final representation was normalized in the FF implementation, MRL models adopted the same for each nested dimension for a fair comparison.

Table 1: Top-1 classification accuracy (%) for ResNet50 MRL and baseline models on ImageNet-1K.

| Rep. Size | Rand. LP | SVD | FF | Slim. Net | MRL | MRL–E |
|---|---|---|---|---|---|---|
| 8 | 4.56 | 2.34 | 65.29 | 0.42 | **66.63** | 56.66 |
| 16 | 11.29 | 7.17 | 72.85 | 0.96 | **73.53** | 71.94 |
| 32 | 27.21 | 20.46 | 74.60 | 2.27 | **75.03** | 74.48 |
| 64 | 49.47 | 48.10 | 75.27 | 5.59 | **75.82** | 75.35 |
| 128 | 65.70 | 67.24 | 75.29 | 14.15 | **76.30** | 75.80 |
| 256 | 72.43 | 74.59 | 75.71 | 38.42 | **76.47** | 76.22 |
| 512 | 74.94 | **76.78** | 76.18 | 69.80 | 76.65 | 76.36 |
| 1024 | 76.10 | **76.87** | 76.63 | 74.61 | 76.76 | 76.48 |
| 2048 | 76.87 | – | **76.87** | 76.26 | 76.80 | 76.51 |

## D   Classification Results

We show the top-1 classification accuracy of ResNet50–MRL models on ImageNet-1K in Table 1 and Figure 2. We compare the performance of MRL models (MRL, MRL–E) to several baselines:

- **FF**: We utilize the FF-k models described in Appendix C for $k \in \{8, ...2048\}$.
- **SVD**: We performed a low rank approximation of the 1000-way classification layer of FF-2048, with rank = 1000.
- **Rand. LP**: We compared against a linear classifier fit on randomly selected features [28].
- **Slim. Net**: We take pretrained slimmable neural networks [95] which are trained with a flexible width backbone (25%, 50%, 75% and full width). For each representation size, we consider the first $k$ dimensions for classification. Note that training of slimmable neural networks becomes unstable when trained below 25% width due to the hardness in optimization and low complexity of the model.

At lower dimensions ( $d \leq 128$), MRL outperforms all baselines significantly, which indicates that pretrained models lack the multifidelity of Matryoshka Representations and are incapable of fitting an accurate linear classifier at low representation sizes.

We compared the performance of MRL models at various representation sizes via 1-nearest neighbors (1-NN) image classification accuracy on ImageNet-1K in Table 2 and Figure 3. We provide detailed information regarding the k-NN search pipeline in Appendix E. We compared against a baseline of attempting to enforce nesting to a FF-2048 model by 1) Random Feature Selection (Rand. FS): considering the first $m$ dimensions of FF-2048 for NN lookup, and 2) FF+SVD: performing SVD on the FF-2048 representations at the specified representation size, 3) FF+JL: performing random projection according to the Johnson-Lindenstrauss lemma [46] on the FF-2048 representations at the specified representation size. We also compared against the 1-NN accuracy of slimmable neural nets [95] as an additional baseline. We observed these baseline models to perform very poorly at lower dimensions, as they were not explicitly trained to learn Matryoshka Representations.

Table 2: 1-NN accuracy (%) on ImageNet-1K for various ResNet50 models.

| Rep. Size | Rand. FS | SVD | JL | FF | Slimmable | MRL | MRL–E |
|---|---|---|---|---|---|---|---|
| 8 | 2.36 | 19.14 | 0.11 | 58.93 | 1.00 | 62.19 | 57.45 |
| 16 | 12.06 | 46.02 | 0.09 | 66.77 | 5.12 | 67.91 | 67.05 |
| 32 | 32.91 | 60.78 | 0.06 | 68.84 | 16.95 | 69.46 | 68.6 |
| 64 | 49.91 | 67.04 | 0.05 | 69.41 | 35.60 | 70.17 | 69.61 |
| 128 | 60.91 | 69.63 | 0.06 | 69.35 | 51.16 | 70.52 | 70.12 |
| 256 | 65.75 | 70.67 | 0.04 | 69.72 | 60.61 | 70.62 | 70.36 |
| 512 | 68.77 | 71.06 | 0.03 | 70.18 | 65.82 | 70.82 | 70.74 |
| 1024 | 70.41 | 71.22 | - | 70.34 | 67.19 | 70.89 | 71.07 |
| 2048 | 71.19 | 71.21 | - | 71.19 | 66.10 | 70.97 | 71.21 |

### D.1   Adaptive Classification (MRL–AC)

In an attempt to use the smallest representation that works well for classification for every image in the ImageNet-1K validation set, we learned a policy to increase the representation size from $m_i$ to

Table 3: Threshold-based adaptive classification performance of ResNet50 MRL on a 40K sized held-out subset of the ImageNet-1K validation set. Results are averaged over 30 random held-out subsets.

| Expected Rep. Size | Accuracy |
|---|---|
| $13.43 \pm 0.81$ | $73.79 \pm 0.10$ |
| $18.32 \pm 1.36$ | $75.25 \pm 0.11$ |
| $25.87 \pm 2.41$ | $76.05 \pm 0.15$ |
| $36.26 \pm 4.78$ | $76.28 \pm 0.16$ |
| $48.00 \pm 8.24$ | $76.43 \pm 0.18$ |
| $64.39 \pm 12.55$ | $76.53 \pm 0.19$ |
| $90.22 \pm 20.88$ | $76.55 \pm 0.20$ |
| $118.85 \pm 33.37$ | $76.56 \pm 0.20$ |

$m_{i+1}$ using a 10K sized subset of the ImageNet-1K validation set. This policy is based on whether the prediction confidence $p_i$ using representation size $m_i$ exceeds a learned threshold $t_i^*$. If $p_i \geq t_i^*$, we used predictions from representation size $m_i$ otherwise, we increased to representation size $m_{i+1}$. To learn the optimal threshold $t_i^*$, we performed a grid search between 0 and 1 (100 samples). For each threshold $t_k$, we computed the classification accuracy over our 10K image subset. We set $t_i^*$ equal to the smallest threshold $t_k$ that gave the best accuracy. We use this procedure to obtain thresholds for successive models, i.e., $\{t_j^* \mid j \in \{8, 16, 32, 64, \ldots, 2048\}\}$. To improve reliability of threshold based greedy policy, we use test time augmentation which has been used successfully in the past [77].

For inference, we used the remaining held-out 40K samples from the ImageNet-1K validation set. We began with smallest sized representation ($m = 8$) and compared the computed prediction confidence $p_8$ to learned optimal threshold $t_8^*$. If $p_8 \leq t_8^*$, then we increased $m = 16$, and repeated this procedure until $m = d = 2048$. To compute the expected dimensions, we performed early stopping at $m = \{16, 32, 64, \ldots 2048\}$ and computed the expectation using the distribution of representation sizes. As shown in Table 3 and Figure 6, we observed that in expectation, we only needed a $\sim 37$ sized representation to achieve $76.3\%$ classification accuracy on ImageNet-1K, which was roughly $14\times$ smaller than the FF–512 baseline. Even if we computed the expectation as a weighted average over the cumulative sum of representation sizes $\{8, 24, 56, \ldots\}$, due to the nature of multiple linear heads for MRL, we ended up with an expected size of 62 that still provided a roughly $8.2\times$ efficient representation than the FF–512 baseline. However, MRL–E alleviates this extra compute with a minimal drop in accuracy.

### D.2 JFT, ALIGN and BERT

We examine the k-NN classification accuracy of learned Matryoshka Representations via ALIGN–MRL and JFT-ViT–MRL in Table 4. For ALIGN [44], we observed that learning Matryoshka Representations via ALIGN–MRL improved classification accuracy at nearly all dimensions when compared to ALIGN. We observed a similar trend when training ViT-B/16 [22] for JFT-300M [80] classification, where learning Matryoshka Representations via MRL and MRL–E on top of JFT-ViT improved classification accuracy for nearly all dimensions, and significantly for lower ones. This demonstrates that training to learn Matryoshka Representations is feasible and extendable even for extremely large scale datasets. We also demonstrate that Matryoshka Representations are learned at interpolated dimensions for both ALIGN and JFT-ViT, as shown in Table 5, despite not being trained explicitly at these dimensions. Lastly, Table 6 shows that MRL training leads to a increase in the cosine similarity span between positive and random image-text pairs.

We also evaluated the capability of Matryoshka Representations to extend to other natural language processing via masked language modeling (MLM) with BERT [19], whose results are tabulated in Table 7. Without any hyper-parameter tuning, we observed Matryoshka Representations to be within $0.5\%$ of FF representations for BERT MLM validation accuracy. This is a promising initial result that could help with large-scale adaptive document retrieval using BERT–MRL.

## E Image Retrieval

We evaluated the strength of Matryoshka Representations via image retrieval on ImageNet-1K (the training distribution), as well as on out-of-domain datasets ImageNetV2 and ImageNet-4K for all

Table 4: ViT-B/16 and ViT-B/16-MRL top-1 and top-5 k-NN accuracy (%) for ALIGN and JFT. Top-1 entries where MRL–E and MRL outperform baselines are bolded for both ALIGN and JFT-ViT.

| Rep. Size | ALIGN | | ALIGN-MRL | | JFT-ViT | | JFT-ViT-MRL | | JFT-ViT-MRL–E | |
|---|---|---|---|---|---|---|---|---|---|---|
| | Top-1 | Top-5 | Top-1 | Top-5 | Top-1 | Top-5 | Top-1 | Top-5 | Top-1 | Top-5 |
| 12 | 11.90 | 28.05 | **43.57** | 67.36 | 27.07 | 48.57 | **53.61** | 75.30 | **51.54** | 73.94 |
| 24 | 33.35 | 55.58 | **56.44** | 78.19 | 48.64 | 70.20 | **62.80** | 81.51 | **62.40** | 81.36 |
| 48 | 51.32 | 73.15 | **62.33** | 82.30 | 63.58 | 81.80 | **67.24** | 84.37 | **66.89** | 83.80 |
| 96 | 61.82 | 81.97 | **65.72** | 84.61 | 68.56 | 85.13 | **69.74** | 85.86 | **68.80** | 85.13 |
| 192 | 66.71 | 85.27 | **67.00** | 85.36 | 71.32 | 86.21 | **71.34** | 86.62 | **70.41** | 86.01 |
| 384 | 67.65 | 85.70 | **67.70** | 85.73 | 71.67 | 86.98 | **71.73** | 87.08 | 71.18 | 86.46 |
| 768 | 68.00 | 86.10 | 67.85 | 85.85 | 72.10 | 87.20 | 71.85 | 86.92 | 71.31 | 86.62 |

Table 5: Examining top-1 and top-5 k-NN accuracy (%) at interpolated hidden dimensions for ALIGN and JFT. This indicates that MRL is able to scale classification accuracy as hidden dimensions increase even at dimensions that were not explicitly considered during training.

| Interpolated Rep. Size | ALIGN-MRL | | JFT-ViT-MRL | |
|---|---|---|---|---|
| | Top-1 | Top-5 | Top-1 | Top-5 |
| 16 | 49.06 | 72.26 | 58.35 | 78.55 |
| 32 | 58.64 | 79.96 | 64.98 | 82.89 |
| 64 | 63.90 | 83.39 | 68.19 | 84.85 |
| 128 | 66.63 | 85.00 | 70.35 | 86.24 |
| 256 | 67.10 | 85.30 | 71.57 | 86.77 |
| 512 | 67.64 | 85.72 | 71.55 | 86.67 |

MRL ResNet50 models. We generated the database and query sets, containing $N$ and $Q$ samples respectively, with a standard PyTorch [63] forward pass on each dataset. We specify the representation size at which we retrieve a shortlist of k-nearest neighbors (k-NN) by $D_s$. The database is a thus a $[N, D_s]$ array, the query set is a $[Q, D_s]$ array, and the neighbors set is a $[Q, k]$ array. For metrics, we utilized corrected mean average precision (mAP@k) [53] and precision (P@k):
$P@k = \dfrac{correct\_pred}{k}$ where $correct\_pred$ is the average number of retrieved NN with the correct label over the entire query set using a shortlist of length $k$.

We performed retrieval with FAISS [45], a library for efficient similarity search. To obtain a shortlist of k-NN, we built an index to search the database. We performed an exhaustive NN search with the L2 distance metric with `faiss.IndexFlatL2`, as well as an approximate NN search (ANNS) via HNSW [45] with `faiss.IndexHNSWFlat`. We used HNSW with $M = 32$ unless otherwise mentioned, and henceforth referred to as HNSW32. The exact search index was moved to the GPU for fast k-NN search computation, whereas the HNSW index was kept on the CPU as it currently lacks GPU support. We show the wall clock times for building the index as well as the index size in Table 20. We observed exact search to have a smaller index size which was faster to build when compared to HNSW, which trades off a larger index footprint for fast NN search (discussed in more detail in Appendix K). The database and query vectors are normalized with `faiss.normalize_L2` before building the index and performing search.

Retrieval performance on ImageNet-1K, *i.e.* the training distribution, is shown in Table 8. MRL outperforms FF models for nearly all representation size for both top-1 and mAP@10, and especially at low representation size ($D_s \leq 32$). MRL–E loses out to FF significantly only at $D_s = 8$. This indicates that training ResNet50 models via the MRL training paradigm improves retrieval at low representation size over models explicitly trained at those representation size (FF-8...2048).

We carried out all retrieval experiments at $D_s \in \{8, 16, 32, 64, 128, 256, 512, 1024, 2048\}$, as these were the representation sizes which were a part of the `nesting_list` at which losses were added during training, as seen in Algorithm 1, Appendix A. To examine whether MRL is able to learn Matryoshka Representations at dimensions in between the representation size for which it was trained, we also tabulate the performance of MRL at interpolated $D_s \in \{12, 24, 48, 96, 192, 384, 768, 1536\}$ as MRL–Interpolated and MRL–E–Interpolated (see Table 8). We observed that performance scaled nearly monotonically between the original representation

Table 6: Cosine similarity between embeddings

| Avg. Cosine Similarity | ALIGN | ALIGN-MRL |
|---|---|---|
| Positive Text to Image | 0.27 | 0.49 |
| Random Text to Image | 8e-3 | -4e-03 |
| Random Image to Image | 0.10 | 0.08 |
| Random Text to Text | 0.22 | 0.07 |

Table 7: Masked Language Modelling (MLM) accuracy(%) of FF and $\mathrm{MRL}$ models on the validation set.

| Rep. Size | BERT-FF | BERT-MRL |
|---|---|---|
| 12 | 60.12 | 59.92 |
| 24 | 62.49 | 62.05 |
| 48 | 63.85 | 63.40 |
| 96 | 64.32 | 64.15 |
| 192 | 64.70 | 64.58 |
| 384 | 65.03 | 64.81 |
| 768 | 65.54 | 65.00 |

size and the interpolated representation size as we increase $D_s$, which demonstrates that $\mathrm{MRL}$ is able to learn $\mathrm{Matryoshka\ Representations}$ at nearly all representation size $m \in [8, 2048]$ despite optimizing only for $|\mathcal{M}|$ nested representation sizes.

We examined the robustness of $\mathrm{MRL}$ for retrieval on out-of-domain datasets ImageNetV2 and ImageNet-4K, as shown in Table 9 and Table 10 respectively. On ImageNetV2, we observed that $\mathrm{MRL}$ outperformed FF at all $D_s$ on top-1 Accuracy and mAP@10, and $\mathrm{MRL{-}E}$ outperformed FF at all $D_s$ except $D_s = 8$. This demonstrates the robustness of the learned $\mathrm{Matryoshka\ Representations}$ for out-of-domain image retrieval.

# F   Adaptive Retrieval

The time complexity of retrieving a shortlist of k-NN often scales as $O(d)$, where $d = D_s$, for a fixed k and $N$. We thus will have a theoretical $256\times$ higher cost for $D_s = 2048$ over $D_s = 8$. We discuss search complexity in more detail in Appendix I. In an attempt to replicate performance at higher $D_s$ while using less FLOPs, we perform adaptive retrieval via retrieving a k-NN shortlist with representation size $D_s$, and then re-ranking the shortlist with representations of size $D_r$. Adaptive retrieval for a shortlist length $k = 200$ is shown in Table 11 for ImageNet-1K, and in Table 12 for ImageNet-4K. On ImageNet-1K, we are able to achieve comparable performance to retrieval with $D_s = 2048$ (from Table 8) with $D_s = 16$ at $128\times$ less MFLOPs/Query (used interchangeably with MFLOPs). Similarly, on ImageNet-4K, we are able to achieve comparable performance to retrieval with $D_s = 2048$ (from Table 10) with $D_s = 64$ on ImageNet-1K and ImageNet-4K, at $32\times$ less MFLOPs. This demonstrates the value of intelligent routing techniques which utilize appropriately sized $\mathrm{Matryoshka\ Representations}$ for retrieval.

Table 8: Retrieve a shortlist of 200-NN with $D_s$ sized representations on ImageNet-1K via exact search with L2 distance metric. Top-1 and mAP@10 entries (%) where MRL–E and MRL outperform FF at their respective representation sizes are bolded.

| Model | $D_s$ | MFlops | Top-1 | Top-5 | Top-10 | mAP@10 | mAP@25 | mAP@50 | mAP@100 | P@10 | P@25 | P@50 | P@100 |
|---|---|---|---|---|---|---|---|---|---|---|---|---|---|
| FF | 8 | 10 | 58.93 | 75.76 | 80.25 | 53.42 | 52.29 | 51.84 | 51.57 | 59.32 | 59.28 | 59.25 | 59.21 |
| | 16 | 20 | 66.77 | 80.88 | 84.40 | 61.63 | 60.51 | 59.98 | 59.62 | 66.76 | 66.58 | 66.43 | 66.27 |
| | 32 | 41 | 68.84 | 82.58 | 86.14 | 63.35 | 62.08 | 61.36 | 60.76 | 68.43 | 68.13 | 67.83 | 67.48 |
| | 64 | 82 | 69.41 | 83.56 | 87.33 | 63.26 | 61.64 | 60.63 | 59.67 | 68.49 | 67.91 | 67.38 | 66.74 |
| | 128 | 164 | 69.35 | 84.23 | 88.24 | 62.30 | 60.16 | 58.73 | 57.29 | 67.84 | 66.83 | 65.96 | 64.92 |
| | 256 | 328 | 69.72 | 84.71 | 88.54 | 61.47 | 58.85 | 57.02 | 55.13 | 67.19 | 65.82 | 64.64 | 63.24 |
| | 512 | 656 | 70.18 | 85.04 | 88.91 | 61.37 | 58.41 | 56.26 | 53.98 | 67.12 | 65.49 | 64.07 | 62.35 |
| | 1024 | 1312 | 70.34 | 85.38 | 89.19 | 61.13 | 57.87 | 55.47 | 52.90 | 66.93 | 65.08 | 63.43 | 61.45 |
| | 2048 | 2624 | 71.19 | 85.66 | 89.17 | 62.90 | 60.06 | 57.99 | 55.76 | 68.46 | 66.9 | 65.52 | 63.83 |
| MRL–E | 8 | 10 | 57.39 | 74.18 | 79.16 | 51.80 | 50.41 | 49.60 | 48.86 | 57.50 | 57.16 | 56.81 | 56.36 |
| | 16 | 20 | **67.08** | 81.38 | 85.15 | 61.60 | 60.36 | 59.66 | 59.04 | 66.79 | 66.53 | 66.24 | 65.87 |
| | 32 | 41 | 68.62 | 82.92 | 86.44 | 63.34 | 61.97 | 61.14 | 60.39 | 68.49 | 68.06 | 67.65 | 67.17 |
| | 64 | 82 | **69.56** | 83.49 | 86.85 | **63.84** | 62.33 | 61.43 | 60.57 | 68.93 | 68.4 | 67.96 | 67.38 |
| | 128 | 164 | **70.13** | 83.63 | 87.07 | **64.15** | 62.58 | 61.61 | 60.70 | 69.19 | 68.62 | 68.11 | 67.50 |
| | 256 | 328 | **70.39** | 83.8 | 87.28 | **64.35** | 62.76 | 61.76 | 60.82 | 69.36 | 68.79 | 68.26 | 67.63 |
| | 512 | 656 | **70.74** | 83.91 | 87.33 | **64.69** | 63.05 | 62.06 | 61.14 | 69.63 | 69.00 | 68.50 | 67.88 |
| | 1024 | 1312 | **71.05** | 84.13 | 87.46 | **64.85** | 63.22 | 62.19 | 61.26 | 69.78 | 69.16 | 68.60 | 67.99 |
| | 2048 | 2624 | 71.17 | 84.27 | 87.67 | **64.99** | 63.33 | 62.29 | 61.33 | 69.90 | 69.24 | 68.68 | 68.05 |
| MRL–E Interpolated | 12 | 15 | 64.25 | 79.21 | 83.29 | 58.83 | 57.50 | 56.71 | 56.02 | 64.10 | 63.78 | 63.42 | 63.02 |
| | 24 | 31 | 68.28 | 82.31 | 85.89 | 62.75 | 61.41 | 60.62 | 59.92 | 67.89 | 67.49 | 67.11 | 66.69 |
| | 48 | 61 | 69.20 | 83.15 | 86.67 | 63.58 | 62.12 | 61.23 | 60.42 | 68.71 | 68.19 | 67.75 | 67.22 |
| | 96 | 123 | 70.05 | 83.63 | 87.11 | 64.04 | 62.46 | 61.52 | 60.63 | 69.10 | 68.51 | 68.04 | 67.45 |
| | 192 | 246 | 70.36 | 83.72 | 87.21 | 64.26 | 62.65 | 61.65 | 60.72 | 69.26 | 68.67 | 68.15 | 67.53 |
| | 384 | 492 | 70.54 | 83.88 | 87.28 | 64.55 | 62.94 | 61.93 | 61.01 | 69.51 | 68.92 | 68.40 | 67.78 |
| | 768 | 984 | 70.96 | 84.05 | 87.44 | 64.79 | 63.15 | 62.15 | 61.22 | 69.72 | 69.10 | 68.56 | 67.95 |
| | 1536 | 1968 | 71.19 | 84.17 | 87.57 | 64.94 | 63.29 | 62.26 | 61.32 | 69.85 | 69.21 | 68.66 | 68.04 |
| MRL | 8 | 10 | **62.19** | 77.05 | 81.34 | **56.74** | 55.47 | 54.76 | 54.12 | 62.06 | 61.81 | 61.54 | 61.17 |
| | 16 | 20 | **67.91** | 81.44 | 85.00 | **62.94** | 61.79 | 61.16 | 60.64 | 67.93 | 67.71 | 67.48 | 67.20 |
| | 32 | 41 | **69.46** | 83.01 | 86.30 | **64.21** | 62.96 | 62.22 | 61.58 | 69.18 | 68.87 | 68.54 | 68.17 |
| | 64 | 82 | **70.17** | 83.53 | 86.95 | **64.69** | 63.33 | 62.53 | 61.80 | 69.67 | 69.25 | 68.89 | 68.42 |
| | 128 | 164 | **70.52** | 83.98 | 87.25 | **64.94** | 63.50 | 62.63 | 61.83 | 69.93 | 69.44 | 69.02 | 68.50 |
| | 256 | 328 | **70.62** | 84.17 | 87.38 | **65.04** | 63.56 | 62.66 | 61.81 | 70.02 | 69.52 | 69.07 | 68.50 |
| | 512 | 656 | **70.82** | 84.31 | 87.55 | **65.14** | 63.57 | 62.62 | 61.73 | 70.12 | 69.53 | 69.04 | 68.45 |
| | 1024 | 1312 | **70.89** | 84.44 | 87.68 | **65.16** | 63.58 | 62.60 | 61.68 | 70.14 | 69.54 | 69.01 | 68.35 |
| | 2048 | 2624 | 70.97 | 84.41 | 87.74 | **65.20** | 63.57 | 62.56 | 61.60 | 70.18 | 69.52 | 68.98 | 68.35 |
| MRL Interpolated | 12 | 15 | 65.89 | 80.04 | 83.68 | 60.84 | 59.66 | 58.98 | 58.37 | 65.94 | 65.72 | 65.45 | 65.08 |
| | 24 | 31 | 68.76 | 82.48 | 85.87 | 63.64 | 62.42 | 61.74 | 61.13 | 68.64 | 68.35 | 68.07 | 67.71 |
| | 48 | 61 | 69.96 | 83.40 | 86.65 | 64.58 | 63.2 | 62.42 | 61.72 | 69.53 | 69.10 | 68.75 | 68.32 |
| | 96 | 123 | 70.40 | 83.83 | 87.04 | 64.86 | 63.46 | 62.62 | 61.84 | 69.82 | 69.38 | 68.98 | 68.48 |
| | 192 | 246 | 70.64 | 84.09 | 87.37 | 65.00 | 63.53 | 62.66 | 61.83 | 69.98 | 69.49 | 69.05 | 68.50 |
| | 384 | 492 | 70.69 | 84.25 | 87.41 | 65.09 | 63.56 | 62.64 | 61.76 | 70.05 | 69.51 | 69.04 | 68.46 |
| | 768 | 984 | 70.84 | 84.40 | 87.63 | 65.16 | 63.59 | 62.62 | 61.71 | 70.14 | 69.55 | 69.03 | 68.44 |
| | 1536 | 1968 | 70.88 | 84.39 | 87.71 | 65.18 | 63.59 | 62.58 | 61.64 | 70.16 | 69.54 | 68.99 | 68.38 |

**Funnel Retrieval.** We also designed a simple cascade policy which we call funnel retrieval to successively improve and refine the k-NN shortlist at increasing $D_s$. This was an attempt to remove the dependence on manual choice of $D_s$ & $D_r$. We retrieved a shortlist at $D_s$ and then re-ranked the shortlist five times while simultaneously increasing $D_r$ (rerank cascade) and decreasing the shortlist length (shortlist cascade), which resembles a funnel structure. We tabulate the performance of funnel retrieval in various configurations in Table 13 on ImageNet-1K, and in Table 14 on ImageNet-4K. With funnel retrieval on ImageNet-1K, we were able to achieve top-1 accuracy within $0.1\%$ of retrieval with $D_s = 2048$ (as in Table 8) with a funnel with $D_s = 16$, with $128\times$ less MFLOPs. Similarly, we are able to achieve equivalent top-1 accuracy within $0.15\%$ of retrieval at $D_s = 2048$ (as in Table 10) with funnel retrieval at $D_s = 32$ on ImageNet-4K, with $64\times$ less MFLOPs. This demonstrates that with funnel retrieval, we can emulate the performance of retrieval with $D_s = 2048$ with a fraction of the MFLOPs.

# G   Few-shot and Sample Efficiency

We compared MRL, MRL–E, and FF on various benchmarks to observe the effect of representation size on sample efficiency. We used Nearest Class Means [74] for classification which has been shown to be effective in the few-shot regime [13].

**ImageNetV2.** Representations are evaluated on ImageNetV2 with the n-shot k-way setup. ImageNetV2 is a dataset traditionally used to evaluate the robustness of models to natural distribution shifts. For our experiments we evaluate accuracy of the model given $n$ examples from the ImageNetV2 distribution. We benchmark representations in the traditional small-scale (10-way) and

Table 9: Retrieve a shortlist of 200-NN with $D_s$ sized representations on ImageNetV2 via exact search with L2 distance metric. Top-1 and mAP@10 entries (%) where MRL–E outperforms FF are bolded. MRL outperforms FF at all $D_s$ and is thus not bolded.

| Config | $D_s$ | MFLOPs | Top-1 | Top-5 | Top-10 | mAP@10 | mAP@25 | mAP@50 | mAP@100 | P@10 | P@25 | P@50 | P@100 |
|---|---|---|---|---|---|---|---|---|---|---|---|---|---|
| FF | 8 | 10 | 48.79 | 64.70 | 69.72 | 43.04 | 41.89 | 41.42 | 41.17 | 48.43 | 48.27 | 48.25 | 48.19 |
|  | 16 | 20 | 55.08 | 69.50 | 74.08 | 49.63 | 48.53 | 48.06 | 47.75 | 54.76 | 54.64 | 54.53 | 54.39 |
|  | 32 | 41 | 56.69 | 71.10 | 76.47 | 51.11 | 49.85 | 49.17 | 48.65 | 56.23 | 55.96 | 55.71 | 55.42 |
|  | 64 | 82 | 57.37 | 72.71 | 77.48 | 51.28 | 49.75 | 48.85 | 47.99 | 56.65 | 56.14 | 55.71 | 55.15 |
|  | 128 | 164 | 57.17 | 73.31 | 78.64 | 50.07 | 48.09 | 46.79 | 45.58 | 55.75 | 54.89 | 54.12 | 53.28 |
|  | 256 | 328 | 57.09 | 74.04 | 79.24 | 49.11 | 46.66 | 44.99 | 43.35 | 55.02 | 53.77 | 52.74 | 51.53 |
|  | 512 | 656 | 57.12 | 73.91 | 79.32 | 48.95 | 46.25 | 44.37 | 42.42 | 54.88 | 53.49 | 52.29 | 50.83 |
|  | 1024 | 1312 | 57.53 | 74.17 | 79.55 | 48.27 | 45.41 | 43.36 | 41.26 | 54.31 | 52.84 | 51.49 | 49.87 |
|  | 2048 | 2624 | 57.84 | 74.59 | 79.45 | 49.99 | 47.47 | 45.66 | 43.87 | 55.89 | 54.63 | 53.45 | 52.12 |
| MRL–E | 8 | 10 | 47.05 | 62.53 | 67.60 | 40.79 | 39.47 | 38.78 | 38.16 | 46.03 | 45.77 | 45.54 | 45.17 |
|  | 16 | 20 | **55.73** | 70.54 | 74.86 | **49.86** | 48.57 | 47.84 | 47.26 | 54.97 | 54.71 | 54.44 | 54.10 |
|  | 32 | 41 | **57.33** | 71.61 | 76.64 | **51.26** | 49.92 | 49.09 | 48.42 | 56.46 | 56.11 | 55.70 | 55.30 |
|  | 64 | 82 | **57.90** | 72.55 | 77.44 | **51.89** | 50.29 | 49.34 | 48.53 | 57.06 | 56.45 | 55.97 | 55.43 |
|  | 128 | 164 | **57.73** | 72.79 | 77.28 | **52.02** | 50.38 | 49.49 | 48.62 | 57.13 | 56.58 | 56.15 | 55.58 |
|  | 256 | 328 | **58.22** | 72.77 | 77.67 | **52.16** | 50.61 | 49.67 | 48.81 | 57.30 | 56.79 | 56.33 | 55.77 |
|  | 512 | 656 | **58.46** | 73.00 | 77.88 | **52.52** | 50.97 | 50.02 | 49.16 | 57.65 | 57.10 | 56.64 | 56.08 |
|  | 1024 | 1312 | **58.71** | 73.29 | 78.00 | **52.70** | 51.13 | 50.17 | 49.30 | 57.83 | 57.26 | 56.77 | 56.20 |
|  | 2048 | 2624 | **58.86** | 73.17 | 78.00 | **52.88** | 51.25 | 50.26 | 49.36 | 57.95 | 57.35 | 56.85 | 56.25 |
| MRL | 8 | 10 | **50.41** | 65.56 | 70.27 | **45.51** | 44.38 | 43.71 | 43.17 | 50.55 | 50.44 | 50.17 | 49.91 |
|  | 16 | 20 | **56.64** | 70.19 | 74.61 | **50.98** | 49.76 | 49.16 | 48.69 | 55.90 | 55.66 | 55.52 | 55.29 |
|  | 32 | 41 | **57.96** | 71.88 | 76.41 | **52.06** | 50.78 | 50.09 | 49.54 | 57.18 | 56.83 | 56.57 | 56.27 |
|  | 64 | 82 | **58.94** | 72.74 | 77.17 | **52.65** | 51.24 | 50.44 | 49.76 | 57.72 | 57.29 | 56.94 | 56.52 |
|  | 128 | 164 | **59.13** | 73.07 | 77.49 | **52.94** | 51.42 | 50.53 | 49.74 | 58.00 | 57.47 | 57.05 | 56.55 |
|  | 256 | 328 | **59.18** | 73.64 | 77.75 | **52.96** | 51.45 | 50.52 | 49.70 | 58.01 | 57.53 | 57.06 | 56.54 |
|  | 512 | 656 | **59.40** | 73.85 | 77.97 | **53.01** | 51.39 | 50.46 | 49.61 | 58.11 | 57.49 | 57.04 | 56.48 |
|  | 1024 | 1312 | **59.11** | 73.77 | 77.92 | **52.98** | 51.37 | 50.40 | 49.54 | 58.13 | 57.51 | 57.00 | 56.45 |
|  | 2048 | 2624 | **59.63** | 73.84 | 77.97 | **52.96** | 51.34 | 50.34 | 49.44 | 58.07 | 57.48 | 56.95 | 56.36 |

Table 10: Retrieve a shortlist of 200-NN with $D_s$ sized representations on ImageNet-4K via exact search with L2 distance metric. MRL–E and FF models are omitted for clarity and compute/inference time costs. All entries are in %.

| Config | $D_s$ | MFLOPs | Top-1 | Top-5 | Top-10 | mAP@10 | mAP@25 | mAP@50 | mAP@100 | P@10 | P@25 | P@50 | P@100 |
|---|---|---|---|---|---|---|---|---|---|---|---|---|---|
| MRL | 8 | 34 | 10.60 | 26.23 | 35.57 | 5.32 | 4.29 | 3.76 | 3.36 | 9.13 | 8.77 | 8.46 | 8.13 |
|  | 16 | 67 | 16.74 | 36.91 | 47.28 | 8.64 | 6.83 | 5.84 | 5.05 | 13.82 | 12.79 | 12.04 | 13.27 |
|  | 32 | 134 | 21.54 | 43.75 | 54.11 | 11.36 | 8.88 | 7.47 | 6.31 | 17.25 | 15.67 | 14.47 | 13.27 |
|  | 64 | 269 | 25.00 | 47.97 | 58.25 | 13.38 | 10.40 | 8.67 | 7.23 | 19.68 | 17.64 | 16.14 | 14.65 |
|  | 128 | 538 | 27.27 | 50.35 | 60.47 | 14.77 | 11.47 | 9.53 | 7.91 | 21.25 | 18.95 | 17.26 | 15.59 |
|  | 256 | 1076 | 28.53 | 51.95 | 61.90 | 15.66 | 12.19 | 10.12 | 8.38 | 22.28 | 19.81 | 18.01 | 16.22 |
|  | 512 | 2151 | 29.46 | 53.03 | 62.81 | 16.29 | 12.70 | 10.55 | 8.72 | 22.96 | 20.42 | 18.54 | 16.68 |
|  | 1024 | 4303 | 30.23 | 53.72 | 63.45 | 16.76 | 13.08 | 10.86 | 8.97 | 23.48 | 20.88 | 18.93 | 17.00 |
|  | 2048 | 8606 | 30.87 | 54.32 | 64.02 | 17.20 | 13.43 | 11.14 | 9.19 | 23.97 | 21.28 | 19.28 | 17.30 |
| MRL-Interpolated | 12 | 50 | 14.04 | 32.56 | 42.71 | 7.16 | 5.70 | 4.92 | 4.32 | 11.81 | 11.08 | 10.52 | 9.94 |
|  | 24 | 101 | 19.49 | 40.82 | 51.26 | 10.17 | 7.98 | 6.75 | 5.75 | 15.76 | 14.43 | 13.42 | 12.40 |
|  | 48 | 202 | 23.51 | 46.23 | 56.56 | 12.49 | 9.72 | 8.13 | 6.81 | 18.62 | 16.75 | 15.39 | 14.04 |
|  | 96 | 403 | 26.25 | 49.32 | 59.48 | 14.15 | 11.00 | 9.15 | 7.61 | 20.55 | 18.36 | 16.78 | 15.17 |
|  | 192 | 807 | 27.94 | 51.32 | 61.32 | 15.29 | 11.89 | 9.88 | 8.18 | 21.86 | 19.46 | 17.71 | 15.96 |
|  | 384 | 1614 | 29.03 | 52.53 | 62.45 | 15.99 | 12.46 | 10.35 | 8.56 | 22.64 | 20.14 | 18.29 | 16.47 |
|  | 768 | 3227 | 29.87 | 53.36 | 63.13 | 16.54 | 12.90 | 10.71 | 8.85 | 23.23 | 20.67 | 18.75 | 16.85 |
|  | 1536 | 6454 | 30.52 | 54.02 | 63.79 | 16.99 | 13.27 | 11.01 | 9.08 | 23.73 | 21.09 | 19.12 | 17.16 |

large-scale (1000-way) setting. We evaluate for $n \in 1, 3, 5, 7, 9$ with 9 being the maximum value for $n$ because there are 10 images per class.

We observed that MRL had equal performance to FF across all representation sizes and shot numbers. We also found that for both MRL and FF, as the shot number decreased, the required representation size to reach optimal accuracy decreased (Table 15). For example, we observed that 1-shot performance at 32 representation size had equal accuracy to 2048 representation size.

**FLUID.** For the long-tailed setting we evaluated MRL on the FLUID benchmark [87] which contains a mixture of pretrain and new classes. Table 16 shows the evaluation of the learned representation on FLUID. We observed that MRL provided up to 2% higher accuracy on novel classes in the tail of the distribution, without sacrificing accuracy on other classes. Additionally we found the accuracy between low-dimensional and high-dimensional representations was marginal for pretrain classes. For example, the 64-dimensional MRL performed $\sim 1\%$ lower in accuracy compared to the 2048-dimensional counterpart on pretrain-head classes (84.46% vs 85.60%). However for novel-tail classes the gap was far larger (6.22% vs 12.88%). We hypothesize that the higher-dimensional representations are required to differentiate the classes when few training examples of each are known.

Table 11: Retrieve a shortlist of k-NN with $D_s$ sized representations on ImageNet-1K with MRL representations, and then re-order the neighbors shortlist with L2 distances using $D_r$ sized representations. Top-1 and mAP@10 entries (%) that are within $0.1\%$ of the maximum value achievable without reranking on MRL representations, as seen in Table 8, are bolded.

| | $D_s$ | $D_r$ | MFLOPs | Top-1 | mAP@10 | mAP@25 | mAP@50 | mAP@100 | P@10 | P@25 | P@50 | P@100 |
|---|---|---|---|---|---|---|---|---|---|---|---|---|
| Shortlist Length = 200 | 8 | 16 | 10 | 68.21 | 63.35 | 62.25 | 61.70 | 61.19 | 68.32 | 68.14 | 67.96 | 67.65 |
| | | 32 | | 69.42 | 64.12 | 62.81 | 62.03 | 61.32 | 69.04 | 68.63 | 68.22 | 67.71 |
| | | 64 | | 70.05 | 64.46 | 63.03 | 62.14 | 61.29 | 69.37 | 68.83 | 68.32 | 67.66 |
| | | 128 | | 70.34 | 64.68 | 63.16 | 62.21 | 61.27 | 69.59 | 68.96 | 68.38 | 67.65 |
| | | 256 | | 70.40 | 64.77 | 63.21 | 62.23 | 61.26 | 69.66 | 69.02 | 68.41 | 67.65 |
| | | 512 | | 70.60 | 64.86 | 63.22 | 62.21 | 61.22 | 69.74 | 69.02 | 68.39 | 67.62 |
| | | 1024 | | 70.71 | 64.88 | 63.23 | 62.20 | 61.20 | 69.76 | 69.01 | 68.39 | 67.60 |
| | | 2048 | | 70.81 | 64.90 | 63.22 | 62.17 | 61.16 | 69.77 | 68.99 | 68.36 | 67.57 |
| | 16 | 32 | 21 | 69.47 | 64.27 | 63.04 | 62.36 | 61.75 | 69.21 | 68.90 | 68.58 | 68.12 |
| | | 64 | | 70.16 | 64.74 | 63.42 | 62.66 | 61.94 | 69.66 | 69.22 | 68.81 | 68.22 |
| | | 128 | | 70.52 | 65.00 | 63.60 | 62.77 | 61.98 | 69.91 | 69.36 | 68.89 | 68.24 |
| | | 256 | | 70.55 | **65.10** | 63.67 | 62.82 | 62.01 | 69.98 | 69.43 | 68.92 | 68.25 |
| | | 512 | | 70.74 | **65.21** | 63.70 | 62.83 | 62.00 | 70.08 | 69.43 | 68.92 | 68.24 |
| | | 1024 | | 70.83 | **65.23** | 63.72 | 62.83 | 61.99 | 70.08 | 69.45 | 68.92 | 68.23 |
| | | 2048 | | **70.90** | **65.27** | 63.73 | 62.82 | 61.97 | 70.10 | 69.44 | 68.90 | 68.21 |
| | 32 | 64 | 41 | 70.16 | 64.69 | 63.35 | 62.57 | 61.93 | 69.68 | 69.26 | 68.92 | 68.51 |
| | | 128 | | 70.52 | 64.97 | 63.54 | 62.73 | 62.04 | 69.95 | 69.47 | 69.06 | 68.59 |
| | | 256 | | 70.63 | 65.07 | 63.63 | 62.79 | 62.07 | 70.04 | 69.55 | 69.12 | 68.61 |
| | | 512 | | 70.82 | **65.17** | 63.66 | 62.80 | 62.06 | 70.11 | 69.57 | 69.12 | 68.60 |
| | | 1024 | | **70.89** | **65.20** | 63.68 | 62.80 | 62.04 | 70.15 | 69.59 | 69.12 | 68.59 |
| | | 2048 | | **70.97** | **65.24** | 63.70 | 62.79 | 62.02 | 70.19 | 69.59 | 69.10 | 68.56 |
| | 64 | 128 | 82 | 70.51 | 64.94 | 63.50 | 62.64 | 61.88 | 69.94 | 69.44 | 69.02 | 68.54 |
| | | 256 | | 70.63 | 65.04 | 63.57 | 62.69 | 61.91 | 70.02 | 69.52 | 69.08 | 68.57 |
| | | 512 | | 70.83 | **65.14** | 63.59 | 62.67 | 61.87 | 70.12 | 69.54 | 69.06 | 68.54 |
| | | 1024 | | **70.89** | **65.16** | 63.59 | 62.65 | 61.85 | 70.15 | 69.54 | 69.05 | 68.52 |
| | | 2048 | | **70.97** | **65.20** | 63.59 | 62.63 | 61.82 | 70.18 | 69.53 | 69.03 | 68.49 |
| | 128 | 256 | 164 | 70.63 | 65.04 | 63.56 | 62.66 | 61.82 | 70.02 | 69.52 | 69.07 | 68.51 |
| | | 512 | | 70.82 | **65.14** | 63.58 | 62.63 | 61.77 | 70.11 | 69.54 | 69.04 | 68.47 |
| | | 1024 | | **70.89** | **65.16** | 63.58 | 62.60 | 61.73 | 70.14 | 69.54 | 69.02 | 68.45 |
| | | 2048 | | **70.97** | **65.20** | 63.57 | 62.57 | 61.68 | 70.18 | 69.52 | 68.99 | 68.41 |
| | 256 | 512 | 328 | 70.82 | **65.14** | 63.57 | 62.62 | 61.74 | 70.12 | 69.53 | 69.04 | 68.45 |
| | | 1024 | | **70.88** | **65.16** | 63.58 | 62.60 | 61.69 | 70.14 | 69.54 | 69.01 | 68.41 |
| | | 2048 | | **70.97** | **65.20** | 63.56 | 62.56 | 61.62 | 70.18 | 69.52 | 68.98 | 68.37 |
| | 512 | 1024 | 656 | **70.90** | **65.16** | 63.58 | 62.60 | 61.68 | 70.14 | 69.54 | 69.01 | 68.41 |
| | | 2048 | | **70.98** | **65.20** | 63.57 | 62.56 | 61.60 | 70.18 | 69.52 | 68.98 | 68.35 |
| | 1024 | 2048 | 1312 | **70.97** | **65.20** | 63.57 | 62.56 | 61.60 | 70.18 | 69.52 | 68.98 | 68.35 |

These results provide further evidence that different tasks require varying capacity based on their difficulty.

## H Robustness Experiments

We evaluated the robustness of MRL models on out-of-domain datasets (ImageNetV2/R/A/Sketch) and compared them to the FF baseline. Each of these datasets is described in Appendix B. The results in Table 17 demonstrate that learning Matryoshka Representations does not hurt out-of-domain generalization relative to FF models, and Matryoshka Representations in fact improve the performance on ImageNet-A. For a ALIGN–MRL model, we examine the the robustness via zero-shot retrieval on out-of-domain datasets, including ObjectNet, in Table 18.

## I In Practice Costs

All approximate NN search experiments via HNSW32 were run on an Intel Xeon 2.20GHz CPU with 24 cores. All exact search experiments were run with CUDA 11.0 on 2xA100-SXM4 NVIDIA GPUs with 40G RAM each.

MRL **models.** As MRL makes minimal modifications to the ResNet50 model in the final fc layer via multiple heads for representations at various scales, it has only an 8MB storage overhead when compared to a standard ResNet50 model. MRL–E has no storage overhead as it has a shared head for logits at the final fc layer.

**Retrieval** Exact search has a search time complexity of $O(dkN)$, and HNSW has a search time complexity of $O(dk \log(N))$, where $N$ is the database size, $d$ is the representation size, and $k$ is the

Table 12: Retrieve a shortlist of k-NN with $D_s$ sized representations on ImageNet-4K with MRL representations, and then re-order the neighbors shortlist with L2 distances using $D_r$ sized representations. Top-1 and mAP@10 entries (%) that are within 0.1% of the maximum value achievable without reranking on MRL representations, as seen in Table 10, are bolded.

| | $D_s$ | $D_r$ | MFLOPs | Top-1 | mAP@10 | mAP@25 | mAP@50 | mAP@100 | P@10 | P@25 | P@50 | P@100 |
|---|---|---|---|---|---|---|---|---|---|---|---|---|
| Shortlist Length = 200 | 8 | 16 | 34 | 16.84 | 8.70 | 6.88 | 5.88 | 5.08 | 13.86 | 12.80 | 11.98 | 11.10 |
| | | 32 | | 20.73 | 10.66 | 8.19 | 6.77 | 5.61 | 16.18 | 14.39 | 13.02 | 11.61 |
| | | 64 | | 23.11 | 11.91 | 9.03 | 7.36 | 6.00 | 17.56 | 15.34 | 13.67 | 11.99 |
| | | 128 | | 24.63 | 12.71 | 9.59 | 7.76 | 6.25 | 18.42 | 15.94 | 14.08 | 12.22 |
| | | 256 | | 25.5 | 13.24 | 9.96 | 8.03 | 6.42 | 19.00 | 16.35 | 14.36 | 12.37 |
| | | 512 | | 26.07 | 13.59 | 10.21 | 8.20 | 6.53 | 19.37 | 16.62 | 14.54 | 12.46 |
| | | 1024 | | 26.52 | 13.85 | 10.40 | 8.34 | 6.61 | 19.65 | 16.80 | 14.68 | 12.53 |
| | | 2048 | | 26.94 | 14.11 | 10.57 | 8.45 | 6.68 | 19.92 | 16.98 | 14.79 | 12.58 |
| | 16 | 32 | 67 | 21.44 | 11.24 | 8.72 | 7.26 | 6.02 | 17.02 | 15.30 | 13.92 | 12.41 |
| | | 64 | | 24.36 | 12.78 | 9.75 | 7.96 | 6.43 | 18.72 | 16.41 | 14.63 | 12.74 |
| | | 128 | | 26.08 | 13.70 | 10.39 | 8.39 | 6.69 | 19.68 | 17.07 | 15.05 | 12.94 |
| | | 256 | | 26.99 | 14.27 | 10.79 | 8.67 | 6.85 | 20.27 | 17.48 | 15.31 | 13.07 |
| | | 512 | | 27.60 | 14.66 | 11.06 | 8.86 | 6.97 | 20.67 | 17.75 | 15.50 | 13.16 |
| | | 1024 | | 28.12 | 14.94 | 11.26 | 8.99 | 7.05 | 20.96 | 17.95 | 15.62 | 13.22 |
| | | 2048 | | 28.56 | 15.21 | 11.43 | 9.11 | 7.12 | 21.23 | 18.13 | 15.73 | 13.27 |
| | 32 | 64 | 134 | 24.99 | 13.35 | 10.35 | 8.59 | 7.09 | 19.61 | 17.52 | 15.92 | 14.21 |
| | | 128 | | 27.17 | 14.61 | 11.27 | 9.26 | 7.51 | 20.99 | 18.52 | 16.62 | 14.59 |
| | | 256 | | 28.33 | 15.37 | 11.83 | 9.67 | 7.77 | 21.80 | 19.12 | 17.05 | 14.81 |
| | | 512 | | 29.12 | 15.88 | 12.20 | 9.94 | 7.93 | 22.33 | 19.51 | 17.32 | 14.94 |
| | | 1024 | | 29.78 | 16.25 | 12.47 | 10.13 | 8.05 | 22.71 | 19.79 | 17.5 | 15.03 |
| | | 2048 | | 30.33 | 16.59 | 12.72 | 10.30 | 8.16 | 23.07 | 20.05 | 17.66 | 15.11 |
| | 64 | 128 | 269 | 27.27 | 14.76 | 11.47 | 9.51 | 7.85 | 21.25 | 18.92 | 17.20 | 15.40 |
| | | 256 | | 28.54 | 15.64 | 12.15 | 10.05 | 8.21 | 22.24 | 19.71 | 17.81 | 15.76 |
| | | 512 | | 29.45 | 16.25 | 12.62 | 10.40 | 8.44 | 22.88 | 20.24 | 18.20 | 15.97 |
| | | 1024 | | 30.19 | 16.69 | 12.96 | 10.66 | 8.60 | 23.35 | 20.61 | 18.46 | 16.10 |
| | | 2048 | | **30.81** | **17.10** | 13.27 | 10.88 | 8.74 | 23.79 | 20.93 | 18.69 | 16.21 |
| | 128 | 256 | 538 | 28.54 | 15.66 | 12.19 | 10.12 | 8.36 | 22.28 | 19.81 | 18.00 | 16.16 |
| | | 512 | | 29.45 | 16.29 | 12.69 | 10.53 | 8.66 | 22.96 | 20.41 | 18.50 | 16.48 |
| | | 1024 | | 30.22 | 16.76 | 13.07 | 10.83 | 8.86 | 23.47 | 20.84 | 18.83 | 16.68 |
| | | 2048 | | **30.86** | **17.19** | 13.41 | 11.09 | 9.03 | 23.95 | 21.22 | 19.12 | 16.84 |
| | 256 | 512 | 1076 | 29.45 | 16.29 | 12.70 | 10.55 | 8.71 | 22.97 | 20.42 | 18.54 | 16.66 |
| | | 1024 | | 30.21 | 16.76 | 13.08 | 10.86 | 8.95 | 23.48 | 20.87 | 18.92 | 16.94 |
| | | 2048 | | **30.85** | **17.20** | 13.43 | 11.14 | 9.15 | 23.97 | 21.27 | 19.26 | 17.16 |
| | 512 | 1024 | 2152 | 30.22 | 16.76 | 13.08 | 10.86 | 8.97 | 23.48 | 20.88 | 18.93 | 17.00 |
| | | 2048 | | **30.87** | **17.20** | 13.43 | 11.14 | 9.19 | 23.97 | 21.28 | 19.28 | 17.28 |
| | 1024 | 2048 | 4303 | **30.87** | **17.20** | 13.43 | 11.15 | 9.19 | 23.97 | 21.28 | 19.28 | 17.29 |

Table 13: Retrieve a shortlist of k-NN with $D_s$ sized representations on ImageNet-1K with MRL. This shortlist is then reranked with funnel retrieval, which uses a rerank cascade with a one-to-one mapping with a monotonically decreasing shortlist length as shown in the shortlist cascade. Top-1 and mAP@10 entries (%) within 0.1% of the maximum achievable without reranking on MRL representations, as seen in Table 8, are bolded.

| $D_s$ | Rerank Cascade | Shortlist Cascade | MFLOPs | Top-1 | Top-5 | Top-10 | mAP@10 | P@10 |
|---|---|---|---|---|---|---|---|---|
| 8 | 16→32→64→128→2048 | 200→100→50→25→10 | 10.28 | 70.22 | 82.63 | 85.49 | 64.06 | 68.65 |
| | | 400→200→50→25→10 | 10.29 | 70.46 | 83.13 | 86.08 | 64.43 | 69.10 |
| | | 800→400→200→50→10 | 10.31 | 70.58 | 83.54 | 86.53 | 64.62 | 69.37 |
| 16 | 32→64→128→256→2048 | 200→100→50→25→10 | 20.54 | **70.90** | 83.96 | 86.85 | **65.19** | 69.97 |
| | | 400→200→50→25→10 | 20.56 | **70.95** | 84.05 | 87.04 | **65.18** | 70.00 |
| | | 800→400→200→50→10 | 20.61 | **70.96** | 84.18 | 87.22 | **65.14** | 70.01 |
| 32 | 64→128→256→512→2048 | 200→100→50→25→10 | 41.07 | **70.96** | 84.32 | 87.47 | **65.21** | 70.11 |
| | | 400→200→50→25→10 | 41.09 | **70.97** | 84.32 | 87.47 | **65.19** | 70.11 |
| | | 800→400→200→50→10 | 41.20 | **70.97** | 84.36 | 87.53 | **65.18** | 70.11 |

shortlist length. To examine real-world performance, we tabulated wall clock search time for every query in the ImageNet-1K and ImageNet-4K validation sets over all representation sizes $d$ in Table 19 for both Exact Search and HNSW32, and ablated wall clock query time over shortlist length $k$ on the ImageNet-1K validation set in Table 21. The wall clock time to build the index and the index size is also shown in Table 20.

Table 14: Retrieve a shortlist of k-NN with $D_s$ sized representations on ImageNet-4K with MRL. This shortlist is then reranked with funnel retrieval, which uses a rerank cascade with a one-to-one mapping with a monotonically decreasing shortlist length as shown in the shortlist cascade. Top-1 and mAP@10 entries (%) within $0.15\%$ of the maximum achievable without reranking on MRL representations, as seen in Table 10, are bolded.

| $D_s$ | Rerank Cascade | Shortlist Cascade | MFLOPs | Top-1 | Top-5 | Top-10 | mAP@10 | P@10 |
|---|---|---|---|---|---|---|---|---|
| 8 | 16→32→64→128→2048 | 200→100→50→25→10 | 33.65 | 26.20 | 46.45 | 54.12 | 12.79 | 17.85 |
| | | 400→200→50→25→10 | 33.66 | 26.55 | 47.02 | 54.72 | 13.02 | 18.15 |
| | | 800→400→200→50→10 | 33.68 | 26.83 | 47.54 | 55.35 | 13.24 | 18.44 |
| 16 | 32→64→128→256→2048 | 200→100→50→25→10 | 67.28 | 29.51 | 51.44 | 59.56 | 15.27 | 21.03 |
| | | 400→200→50→25→10 | 67.29 | 29.66 | 51.71 | 59.88 | 15.42 | 21.22 |
| | | 800→400→200→50→10 | 67.34 | 29.79 | 52.00 | 60.25 | 15.55 | 21.41 |
| 32 | 64→128→256→512→2048 | 200→100→50→25→10 | 134.54 | 30.64 | 53.52 | 62.16 | 16.45 | 22.64 |
| | | 400→200→50→25→10 | 134.56 | 30.69 | 53.65 | 62.31 | 16.51 | 22.73 |
| | | 800→400→200→50→10 | 134.66 | **30.72** | 53.78 | 62.43 | 16.55 | 22.79 |
| 64 | 128→256→512→1024→2048 | 200→100→50→25→10 | 269.05 | **30.81** | 54.06 | 63.15 | 16.87 | 23.34 |
| | | 400→200→50→25→10 | 269.10 | **30.84** | 54.20 | 63.31 | 16.92 | 23.42 |
| | | 800→400→200→50→10 | 269.31 | **30.87** | 54.27 | 63.42 | 16.95 | 23.46 |

Table 15: Few-shot accuracy (%) on ImageNetV2 for 1000-way classification. MRL performs equally to FF across all shots and representation sizes. We also observed that accuracy saturated at a lower dimension for lower shot numbers. E.g. for 1-shot, 32-dim performed comparably to 2048-dim.

| Rep. Size | Method | 1-Shot | 3-Shot | 5-Shot | 7-Shot | 9-Shot |
|---|---|---|---|---|---|---|
| 8 | FF | 35.41 | 45.73 | 49.23 | 50.89 | 51.72 |
| | MRL | 35.37 | 45.69 | 49.25 | 50.85 | 51.73 |
| 16 | FF | 40.88 | 53.96 | 57.36 | 58.72 | 59.39 |
| | MRL | 40.90 | 53.94 | 57.37 | 58.65 | 59.29 |
| 32 | FF | 41.41 | 54.88 | 58.28 | 59.63 | 60.40 |
| | MRL | 41.40 | 54.91 | 58.30 | 59.65 | 60.45 |
| 64 | FF | 41.25 | 54.83 | 58.29 | 59.82 | 60.61 |
| | MRL | 41.28 | 54.80 | 58.32 | 59.77 | 60.69 |
| 128 | FF | 41.36 | 54.90 | 58.50 | 60.05 | 60.90 |
| | MRL | 41.38 | 54.95 | 58.50 | 60.06 | 60.83 |
| 256 | FF | 41.36 | 54.90 | 58.50 | 60.05 | 60.90 |
| | MRL | 41.38 | 54.95 | 58.50 | 60.06 | 60.83 |
| 512 | FF | 41.36 | 55.05 | 58.70 | 60.19 | 61.02 |
| | MRL | 41.34 | 55.14 | 58.78 | 60.40 | 61.18 |
| 1024 | FF | 41.32 | 55.20 | 58.85 | 60.46 | 61.38 |
| | MRL | 41.31 | 55.24 | 58.86 | 60.42 | 61.34 |
| 2048 | FF | 41.18 | 55.09 | 58.77 | 60.38 | 61.34 |
| | MRL | 41.16 | 55.10 | 58.77 | 60.40 | 61.28 |

## J  Analysis of Model Disagreement

**Class Trends**  *Does increasing representation size necessarily help improve classification performance across all classes in ImageNet-1K?* We studied this question by examining trends in performance with increasing representation size from $d = 8, ...2048$. For MRL models, we observed that 244 classes showed a monotonic improvement in performance with increasing $d$, 177 classes first improved but then observed a slight dip (one or two misclassifications per class), 49 classes showed a decline first and then an improvement, and the remaining classes did not show a clear trend. When we repeated this experiment with independently trained FF models, we noticed that 950 classes did not show a clear trend. This motivated us to leverage the disagreement as well as gradual improvement of accuracy at different representation sizes by training Matryoshka Representations. Figure 12 showcases the progression of relative per-class accuracy distribution compared to the

Table 16: Accuracy (%) categories indicates whether classes were present during ImageNet pretraining and head/tail indicates classes that have greater/less than 50 examples in the streaming test set. We observed that MRL performed better than the baseline on novel tail classes by $\sim 2\%$ on average.

| Rep. Size | Method | Pretrain - Head (>50) | Novel - Head (>50) | Pretrain - Tail (<50) | Novel - Tail (<50) | Mean Per Class Acc. | Acc. |
|---|---|---|---|---|---|---|---|
| 8 | FF | 68.04 | **11.30** | 33.18 | **0.36** | 16.29 | 28.47 |
| | MRL | **71.75** | 10.70 | **38.29** | 0.19 | **17.15** | **29.34** |
| | MRL–E | 57.40 | 6.25 | 23.14 | 0.04 | 11.78 | 22.81 |
| 16 | FF | 80.74 | **19.12** | **63.29** | **2.78** | **25.65** | **37.61** |
| | MRL | **81.79** | 17.90 | 61.39 | 1.95 | 24.73 | 37.59 |
| | MRL–E | 79.08 | 9.15 | 60.33 | 0.08 | 20.45 | 30.24 |
| 32 | FF | **83.67** | **24.30** | 66.66 | **4.23** | **28.86** | **42.40** |
| | MRL | 83.46 | 23.26 | 65.82 | 3.75 | 28.16 | 41.90 |
| | MRL–E | 81.42 | 10.47 | 68.01 | 0.23 | 22.31 | 32.17 |
| 64 | FF | 84.12 | 27.49 | 68.20 | 5.17 | 30.64 | 45.18 |
| | MRL | **84.46** | **27.61** | 67.59 | **6.22** | **31.03** | **45.35** |
| | MRL–E | 82.57 | 13.23 | **70.18** | 0.52 | 23.83 | 34.74 |
| 128 | FF | 84.87 | 29.96 | **68.79** | 5.54 | 31.84 | 47.06 |
| | MRL | **84.88** | **30.86** | 68.58 | **8.41** | **33.23** | **47.79** |
| | MRL–E | 82.76 | 18.93 | 64.46 | 2.22 | 25.75 | 39.19 |
| 256 | FF | 84.77 | 32.78 | **69.96** | 7.21 | 33.65 | 49.15 |
| | MRL | **85.10** | **32.91** | 69.39 | **9.99** | **34.74** | **49.39** |
| | MRL–E | 82.96 | 22.63 | 64.55 | 3.59 | 27.64 | 41.96 |
| 512 | FF | **85.62** | **35.27** | **70.27** | 9.05 | 35.42 | **51.14** |
| | MRL | **85.62** | 34.67 | 70.24 | **11.43** | **36.11** | 50.79 |
| | MRL–E | 82.86 | 25.62 | 64.34 | 4.99 | 29.22 | 44.20 |
| 1024 | FF | **86.30** | 37.49 | **71.12** | 10.92 | **37.14** | **52.88** |
| | MRL | 85.64 | **35.88** | 70.02 | **12.19** | 36.80 | 51.58 |
| | MRL–E | 83.03 | 27.78 | 64.58 | 6.32 | 30.57 | 45.71 |
| 2048 | FF | **86.40** | **37.09** | **71.74** | 10.77 | 37.04 | **52.67** |
| | MRL | 85.60 | 36.83 | 70.34 | **12.88** | **37.46** | 52.18 |
| | MRL–E | 83.01 | 29.99 | 65.37 | 7.60 | 31.97 | 47.16 |

Table 17: Top-1 classification accuracy (%) on out-of-domain datasets (ImageNet-V2/R/A/Sketch) to examine robustness of Matryoshka Representation Learning. Note that these results are without any fine tuning on these datasets.

| | ImageNet-V1 | | | ImageNet-V2 | | | ImageNet-R | | | ImageNet-A | | | ImageNet-Sketch | | |
|---|---|---|---|---|---|---|---|---|---|---|---|---|---|---|---|
| Rep. Size | FF | MRL–E | MRL | FF | MRL–E | MRL | FF | MRL–E | MRL | FF | MRL–E | MRL | FF | MRL–E | MRL |
| 8 | 65.86 | 56.92 | 67.46 | 54.05 | 47.40 | 55.59 | 24.60 | 22.98 | 23.57 | 2.92 | 3.63 | 3.39 | 17.73 | 15.07 | 17.98 |
| 16 | 73.10 | 72.38 | 73.80 | 60.52 | 60.48 | 61.71 | 28.51 | 28.45 | 28.85 | 3.00 | 3.55 | 3.59 | 21.70 | 20.38 | 21.77 |
| 32 | 74.68 | 74.80 | 75.26 | 62.24 | 62.23 | 63.05 | 31.28 | 30.79 | 31.47 | 2.60 | 3.65 | 3.57 | 22.03 | 21.87 | 22.48 |
| 64 | 75.45 | 75.48 | 76.17 | 63.51 | 63.15 | 63.99 | 32.96 | 32.13 | 33.39 | 2.87 | 3.99 | 3.76 | 22.13 | 22.56 | 23.43 |
| 128 | 75.47 | 76.05 | 76.46 | 63.67 | 63.52 | 64.69 | 33.93 | 33.48 | 34.54 | 2.81 | 3.71 | 3.73 | 22.73 | 22.73 | 23.70 |
| 256 | 75.78 | 76.31 | 76.66 | 64.13 | 63.80 | 64.71 | 34.80 | 33.91 | 34.85 | 2.77 | 3.65 | 3.60 | 22.63 | 22.88 | 23.59 |
| 512 | 76.30 | 76.48 | 76.82 | 64.11 | 64.09 | 64.78 | 35.53 | 34.20 | 34.97 | 2.37 | 3.57 | 3.59 | 23.41 | 22.89 | 23.67 |
| 1024 | 76.74 | 76.60 | 76.93 | 64.43 | 64.20 | 64.95 | 36.06 | 34.22 | 34.99 | 2.53 | 3.56 | 3.68 | 23.44 | 22.98 | 23.72 |
| 2048 | 77.10 | 76.65 | 76.95 | 64.69 | 64.17 | 64.93 | 37.10 | 34.29 | 35.07 | 2.93 | 3.49 | 3.59 | 24.05 | 23.01 | 23.70 |

Matryoshka Representation Learning-2048 dimensional model. This also showed that some instances and classes could benefit from lower-dimensional representations.

**Discussion of Oracle Accuracy**    Based on our observed model disagreements for different representation sizes $d$, we defined an optimal *oracle* accuracy [56] for MRL. We labeled an image as correctly predicted if classification using any representation size was correct. The percentage of total samples of ImageNet-1K that were firstly correctly predicted using each representation size $d$ is shown in Table 22. This defined an upper bound on the performance of MRL models, as $18.46\%$ of the ImageNet-1K validation set were incorrectly predicted $\forall d \in \{8, 16, \ldots, 2048\}$. We show the oracle performance on MRL models for ImageNet-1K/V2/A/R/Sketch datasets in Table 23.

In an attempt to derive an optimal routing policy to emulate oracle accuracy, we designed the adaptive classification via cascading method as discussed in Appendix D.1. This led to an interesting

Table 18: Zero-shot top-1 image classification accuracy (%) of a ALIGN-MRL model on ImageNet-V1/V2/R/A and ObjectNet.

| Rep. Size | V1 | V2 | A | R | ObjectNet |
|---|---|---|---|---|---|
| 12 | 30.57 | 23.98 | 14.59 | 24.24 | 25.52 |
| 24 | 45.64 | 37.71 | 22.75 | 46.40 | 35.89 |
| 48 | 53.84 | 46.16 | 28.88 | 60.71 | 42.76 |
| 96 | 58.31 | 51.34 | 33.21 | 70.12 | 45.20 |
| 192 | 60.95 | 53.56 | 36.10 | 74.41 | 48.24 |
| 384 | 62.06 | 54.77 | 37.95 | 76.51 | 49.10 |
| 768 | 62.26 | 55.15 | 37.84 | 76.73 | 49.26 |
| Baseline | 66.39 | 59.57 | 39.97 | 80.49 | 51.60 |

Table 19: Retrieval k-NN wall clock search times (s) over the entire validation (query) set of ImageNet-1K and ImageNet-4K, containing 50K and 200K samples respectively.

| Rep. Size | ImageNet-1K | | ImageNet-4K | |
|---|---|---|---|---|
| | ExactL2 | HNSW32 | ExactL2 | HNSW32 |
| 8 | 0.60 | 0.14 | 35.70 | 1.17 |
| 16 | 0.57 | 0.18 | 36.16 | 1.65 |
| 32 | 0.60 | 0.20 | 36.77 | 1.75 |
| 64 | 0.66 | 0.24 | 27.88 | 2.21 |
| 128 | 0.86 | 0.32 | 30.10 | 4.15 |
| 256 | 1.29 | 0.46 | 34.97 | 3.39 |
| 512 | 2.17 | 0.68 | 46.97 | 4.83 |
| 1024 | 3.89 | 1.05 | 70.59 | 7.14 |
| 2048 | 7.31 | 2.05 | 117.78 | 13.43 |

Table 20: FAISS [45] index size and build times for exact k-NN search with L2 Distance metric and approximate k-NN search with HNSW32 [59].

| | Exact Search | | | | HNSW32 | | | |
|---|---|---|---|---|---|---|---|---|
| Rep. Size | ImageNet-1K | | ImageNet-4K | | ImageNet-1K | | ImageNet-4K | |
| | Index Size (MB) | Index Build Time (s) | Index Size (MB) | Index Build Time (s) | Index Size (MB) | Index Build Time (s) | Index Size (MB) | Index Build Time (s) |
| 8 | 40 | 0.04 | 131 | 0.33 | 381 | 4.87 | 1248 | 24.04 |
| 16 | 80 | 0.08 | 263 | 0.27 | 421 | 6.15 | 1379 | 33.31 |
| 32 | 160 | 0.16 | 525 | 0.52 | 501 | 6.80 | 1642 | 37.41 |
| 64 | 320 | 0.38 | 1051 | 1.05 | 661 | 8.31 | 2167 | 47.23 |
| 128 | 641 | 0.64 | 2101 | 2.10 | 981 | 11.73 | 3218 | 89.87 |
| 256 | 1281 | 1.27 | 4202 | 4.20 | 1622 | 17.70 | 5319 | 102.84 |
| 512 | 2562 | 2.52 | 8404 | 8.39 | 2903 | 27.95 | 9521 | 158.47 |
| 1024 | 5125 | 5.10 | 16808 | 17.20 | 5465 | 44.02 | 17925 | 236.30 |
| 2048 | 10249 | 10.36 | 33616 | 41.05 | 10590 | 86.15 | 34733 | 468.18 |

Table 21: Retrieval k-NN wall clock search times (s) over entire validation (query) set of ImageNet-1K over various shortlist lengths $k$.

| Index | k = 50 | k = 100 | k = 200 | k = 500 | k = 1000 | k = 2048 |
|---|---|---|---|---|---|---|
| Exact L2 | 0.4406 | 0.4605 | 0.5736 | 0.6060 | 1.2781 | 2.7047 |
| HNSW32 | 0.1193 | 0.1455 | 0.1833 | 0.2145 | 0.2333 | 0.2670 |

observation on the expected dimensionality for 76.30% top-1 classification accuracy being just $d \sim 37$. We leave the design and learning of a more optimal policy for future work.

**Grad-CAM Examples** We analyzed the nature of model disagreement across representation sizes with MRL models with the help of Grad-CAM visualization [75]. We observed there were certain classes in ImageNet-1K such as "tools", "vegetables" and "meat cutting knife" which were occasionally located around multiple objects and a cluttered environment. In such scenarios, we observed that smaller representation size models would often get confused due to other objects and fail to extract the object of interest which generated the correct label. We also observed a different nature

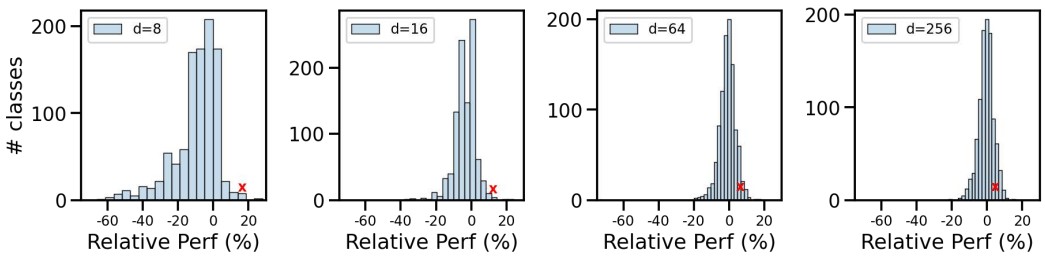

Figure 12: Progression of relative per-class accuracy vs MRL-2048. As the dimensionality increases, the spread shrinks while the class marked (**x**) (Madagascar cat) loses accuracy.

Table 22: Percentage of ImageNet-1K validation set that is first correctly predicted using each representation size $d$. We note that $18.46\%$ of the samples cannot be correctly predicted by any representation size. The remaining $81.54\%$ constitutes the oracle accuracy.

| Rep. Size | 8 | 16 | 32 | 64 | 128 | 256 | 512 | 1024 | 2048 | Always Wrong |
|---|---|---|---|---|---|---|---|---|---|---|
| Correctly Predicted | 67.46 | 8.78 | 2.58 | 1.35 | 0.64 | 0.31 | 0.20 | 0.12 | 0.06 | 18.46 |

of disagreement arising when the models got confused within the same superclass. For example, ImageNet-1K has multiple "snake" classes, and models often confuse a snake image for an incorrect species of snake.

**Superclass Performance**   We created a 30 superclass subset of the validation set based on wordnet hierarchy (Table 24) to quantify the performance of MRL model on ImageNet-1K superclasses. Table 25 quantifies the performance with different representation size.

# K   Ablation Studies

## K.1   MRL **Training Paradigm**

Matryoshka Representations **via Finetuning.**   To observe if nesting can be induced in models that were not explicitly trained with nesting from scratch, we loaded a pretrained FF-2048 ResNet50 model and initialized a new MRL layer, as defined in Algorithm 2, Appendix C. We then unfroze different layers of the backbone to observe how much non-linearity in the form of unfrozen conv layers needed to be present to enforce nesting into a pretrained FF model. A description of these layers can be found in the ResNet50 architecture [27]. All models were finetuned with the FFCV pipeline, with same training configuration as in the end-to-end training aside from changing lr $= 0.1$ and epochs $= 10$. We observed that finetuning the linear layer alone was insufficient to learn Matryoshka Representations at lower dimensionalities. Adding more and more non-linear conv+ReLU layers steadily improved classification accuracy of $d = 8$ from $5\%$ to $60\%$ after finetuning, which was only $6\%$ less than training MRL end-to-end for 40 epochs. This difference was successively less pronounced as we increased dimensionality past $d = 64$, to within $1.5\%$ for all larger dimensionalities. The full results of this ablation can be seen in Table 26.

**Relative Importance.**   We performed an ablation of MRL over the relative importance, $c_m$, of different nesting dimensions $m \in \mathcal{M}$, as defined in Sec. 3. In an attempt to improve performance at lower dimensionalities, we boosted the relative importance $c_m$ of training loss at lower dimensions as in Eq. 1 with two models, MRL-8boost and MRL-8+16boost. The MRL-8boost model had $c_{m \in \mathcal{M}} = [2, 1, 1, 1, 1, 1, 1, 1, 1]$ and the MRL-8+16boost model had $c_{m \in \mathcal{M}} = [2, 1.5, 1, 1, 1, 1, 1, 1, 1]$. The relative importance list $c_{m \in \mathcal{M}}$ had a 1-to-1 correspondence with nesting dimension set $\mathcal{M}$. In Table 27, we observed that MRL-8boost improves top-1 accuracy by $3\%$ at $d = 8$, and also improves top-1 accuracy of all representation scales from 16 to 256 over MRL, while only hurting the performance at 512 to 2048 representation scales by a maximum of $0.1\%$. This suggests that the relative importance $c_m$ can be tuned/set for optimal accuracy for all $m \in \mathcal{M}$, but we leave this extension for future work.

Table 23: Oracle classification accuracy of various evaluation datasets for ResNet50–MRL model trained on ImageNet-1K.

| Top-1 | ImageNetV1 | ImageNetV2 | ImageNet-A | ImageNet-R | ImageNet-Sketch |
|---|---|---|---|---|---|
| FF–2048 | 76.9 | 64.9 | 3.6 | 35.1 | 23.7 |
| MRL–Oracle | 81.5 | 70.6 | 8.7 | 39.8 | 28.9 |

Table 24: 30 Superclasses in ImageNet-1K corresponding to the performance in Table 25.

| | | | | |
|---|---|---|---|---|
| insect | motor vehicle | artiodactyl | vegetable | game equipment |
| terrier | serpent | machine | measuring device | sheepdog |
| protective covering | sporting dog | vessel, watercraft | building | lizard |
| garment | hound | monkey | home appliance | wind instrument |
| vessel | fish | nourishment | electronic equipment | oscine |
| furniture | wading bird | tool | canine | mechanism |

Table 25: Performance of MRL model on 31-way classification (1 extra class is for reject token) on ImageNet-1K superclasses.

| Rep. Size | 8 | 16 | 32 | 64 | 128 | 256 | 512 | 1024 | 2048 |
|---|---|---|---|---|---|---|---|---|---|
| MRL | 85.57 | 88.67 | 89.48 | 89.82 | 89.97 | 90.11 | 90.18 | 90.22 | 90.21 |

Matryoshka Representations **at Arbitrary Granularities.** To train MRL, we used nested dimensions at logarithmic granularities $\mathcal{M} = \{8, 16, \ldots, 1024, 2048\}$ as detailed in Section 3. We made this choice for two empirically-driven reasons: a) The accuracy improvement with increasing representation size was more logarithmic than linear (as shown by FF models in Figure 2). This indicated that optimizing for granularities increasing in a non-logarithmic fashion would be sub-optimal both for maximum performance and expected efficiency; b) If we have $m$ arbitrary granularities, the expected cost of the linear classifier to train MRL scales as $O(L * (m^2))$ while logarithmic granularities result in $O(L * 2log(d))$ space and compute costs.

To demonstrate this effect, we learned Matryoshka Representations with uniform (MRL-*Uniform*) nesting dimensions $m \in \mathcal{M} = \{8, 212, 416, 620, 824, 1028, 1232, 1436, 1640, 1844, 2048\}$. We evaluated this model at the standard (MRL-*log*) dimensions $m \in \mathcal{M} = \{8, 16, 32, 64, 128, 256, 512, 1024, 2048\}$ for ease of comparison to reported numbers using 1-NN accuracy (%). As shown in Table 29, we observed that while performance interpolated, MRL-*Uniform* suffered at low dimensions as the logarithmic spacing of MRL-*log* resulted in tighter packing of information in these initial dimensions. The higher nesting dimensions of MRL-*Uniform* did not help in significant accuracy improvement due to accuracy saturation, which is often logarithmic in representation size as shown by FF models. Note that the slight improvement at dimensions higher than 512 for MRL-*Uniform* is due to multiple granularities around them compared to just three for MRL-*log*, which are not useful in practice for efficiency.

**Lower Dimensionality.** We experimented with training MRL with smaller nesting dimension than $m = 8$, as shown in Table 28, with two models: MRL-4 and MRL-6. We found that using lower than 8-dimensions to train MRL, i.e. $m_0 \in \{4, 6\}$ for MRL-4 and MRL-6 respectively, did not affect the top-1 accuracy of other granularities significantly. However, granularities smaller than 8-dimensions had very low accuracy and were often unusable for deployment along with additional training difficulty. We also observed a small dip in accuracy at higher dimensions which we attribute to the joint loss that now also included the harder optimization of the smallest dimension. Lastly, we hypothesize the dimensionality of 8 is an empirically validated design choice due to the considerable accuracy it provided along with the ease of training.

## K.2 Retrieval

**Adaptive Retrieval.** To examine the effect of increasing shortlist lengths on search time, we performed a reranking ablation over shortlist lengths for $D_s = 16$ and $D_r = 2048$ over ImageNet-1K in Table 30, and over ImageNet-4K in Table 31. We observed that using a larger shortlist $k$ saturated ImageNet-1K performance at $k=200$. But using larger shortlists until $k = 2048$, the maximum value

Table 26: Top-1 classification accuracy (%) on ImageNet-1K of various ResNet50 models which are finetuned on pretrained FF-2048 model. We observed that adding more non-linearities is able to induce nesting to a reasonable extent even if the model was not pretrained with nesting in mind.

| Rep. Size | fc | 4.2 conv3, fc | 4.2 conv2, conv3, fc | 4.2 full, fc | All (MRL) |
|---|---|---|---|---|---|
| 8 | 5.15 | 36.11 | 54.78 | 60.02 | 66.63 |
| 16 | 13.79 | 58.42 | 67.26 | 70.10 | 73.53 |
| 32 | 32.52 | 67.81 | 71.62 | 72.84 | 75.03 |
| 64 | 52.66 | 72.42 | 73.61 | 74.29 | 75.82 |
| 128 | 64.60 | 74.41 | 74.67 | 75.03 | 76.30 |
| 256 | 69.29 | 75.30 | 75.23 | 75.38 | 76.47 |
| 512 | 70.51 | 75.96 | 75.47 | 75.64 | 76.65 |
| 1024 | 70.19 | 76.18 | 75.70 | 75.75 | 76.76 |
| 2048 | 69.72 | 76.44 | 75.96 | 75.97 | 76.80 |

Table 27: An ablation over boosting training loss at lower nesting dimensions, with top-1 and top-5 accuracy (%). The models are described in Appendix K.1.

| Model | MRL | | MRL-8boost | | MRL-8+16boost | |
|---|---|---|---|---|---|---|
| Rep. Size | Top-1 | Top-5 | Top-1 | Top-5 | Top-1 | Top-5 |
| 8 | 66.63 | 84.66 | **69.53** | 86.19 | 69.24 | 85.96 |
| 16 | 73.53 | 89.52 | 73.86 | 89.44 | **73.91** | 89.55 |
| 32 | 75.03 | 91.31 | **75.28** | 91.21 | 75.10 | 91.14 |
| 64 | 75.82 | 92.27 | **75.84** | 92.22 | 75.67 | 92.06 |
| 128 | **76.30** | 92.82 | 76.28 | 92.74 | 76.07 | 92.52 |
| 256 | 76.47 | 93.02 | **76.48** | 92.97 | 76.22 | 92.72 |
| 512 | **76.65** | 93.13 | 76.56 | 93.09 | 76.35 | 92.85 |
| 1024 | **76.76** | 93.22 | 76.71 | 93.21 | 76.39 | 92.98 |
| 2048 | **76.80** | 93.32 | 76.76 | 93.28 | 76.52 | 93.05 |

Table 28: An ablation over training with smaller nesting dimensionalities in terms of Top-1 accuracy (%). MRL-4 and MRL-6 are variations of the original model (MRL-8) with $m_0 \in \{4, 6\}$, where $m \in \mathcal{M}$ is part of the nesting_list as seen in Alg 2.

| Rep. Size | MRL-4 | MRL-6 | MRL-8 |
|---|---|---|---|
| 4 | 27.25 | - | - |
| 6 | - | 58.71 | - |
| 8 | 66.86 | **67.55** | 66.63 |
| 16 | 73.36 | 73.10 | **73.53** |
| 32 | 74.82 | 74.49 | **75.03** |
| 64 | 75.51 | 75.32 | **75.82** |
| 128 | 75.93 | 75.61 | **76.30** |
| 256 | 76.08 | 75.82 | **76.47** |
| 512 | 76.31 | 75.93 | **76.65** |
| 1024 | 76.38 | 76.04 | **76.76** |
| 2048 | 76.43 | 76.12 | **76.80** |

Table 29: An ablation over training MRL with nesting list at uniformly distributed granularities. Entries in the MRL-Uniform column are evaluated at logarithmic dimensions for a fair comparison to MRL-Log (standard MRL) with 1-NN accuracy (%).

| Rep. Size | MRL-Log | MRL-Uniform |
|---|---|---|
| 8 | **62.19** | 58.44 |
| 16 | **67.91** | 61.11 |
| 32 | **69.46** | 63.82 |
| 64 | **70.17** | 66.44 |
| 128 | **70.52** | 68.71 |
| 256 | **70.62** | 70.06 |
| 512 | 70.82 | **70.98** |
| 1024 | 70.89 | **71.37** |
| 2048 | 70.97 | **71.44** |

supported by the FAISS framework, steadily improved performance on ImageNet-4K. This is likely due to the increased database size, but could also indicate a correlation with ImageNet-4K being slightly out-of-distribution making the task at hand harder.

Table 30: Adaptive retrieval ablation over shortlist length $k$ for $D_s = 16$, $D_r = 2048$ on ImageNet-1K with exact search. Entries with the highest P@1 and mAP@10 across all $k$ are in bold.

| Shortlist Length | P@1 | mAP@10 | mAP@25 | mAP@50 | mAP@100 | P@10 | P@25 | P@50 | P@100 |
|---|---|---|---|---|---|---|---|---|---|
| 100 | 70.88 | 65.19 | 63.62 | 62.59 | 61.24 | 69.96 | 69.24 | 68.53 | 67.20 |
| 200 | 70.90 | **65.27** | 63.73 | 62.82 | 61.97 | 70.10 | 69.44 | 68.90 | 68.21 |
| 400 | 70.94 | 65.26 | 63.71 | 62.81 | 62.03 | 70.15 | 69.51 | 69.02 | 68.47 |
| 800 | 70.96 | 65.23 | 63.64 | 62.69 | 61.85 | 70.16 | 69.52 | 69.02 | 68.45 |
| 1600 | 70.96 | 65.20 | 63.58 | 62.58 | 61.66 | 70.16 | 69.5 | 68.97 | 68.36 |
| 2048 | **70.97** | 65.20 | 63.57 | 62.58 | 61.64 | 70.16 | 69.5 | 68.97 | 68.35 |

Table 31: Adaptive retrieval ablation over shortlist length $k$ for $D_s = 16$, $D_r = 2048$ on ImageNet-4K with exact search.

| Shortlist Length | P@1 | mAP@10 | mAP@25 | mAP@50 | mAP@100 | P@10 | P@25 | P@50 | P@100 |
|---|---|---|---|---|---|---|---|---|---|
| 100 | 27.70 | 14.38 | 10.62 | 8.26 | 6.07 | 20.12 | 16.87 | 14.29 | 11.26 |
| 200 | 28.56 | 15.21 | 11.43 | 9.11 | 7.12 | 21.23 | 18.13 | 15.73 | 13.27 |
| 400 | 29.34 | 15.83 | 12.06 | 9.76 | 7.79 | 22.08 | 19.09 | 16.83 | 14.54 |
| 800 | 29.86 | 16.30 | 12.53 | 10.23 | 8.26 | 22.72 | 19.83 | 17.65 | 15.45 |
| 1600 | 30.24 | 16.63 | 12.86 | 10.56 | 8.60 | 23.18 | 20.36 | 18.23 | 16.11 |
| 2048 | **30.35** | **16.73** | 12.96 | 10.65 | 8.69 | 23.31 | 20.50 | 18.40 | 16.30 |