# OpenReview forum: "Matryoshka Representation Learning"
_NeurIPS.cc/2022/Conference — NeurIPS 2022 Accept_

### Official Review · Reviewer_GFsE · 2022-07-05

**Rating:** 7
**Confidence:** 3
**Soundness:** 3 good
**Presentation:** 3 good
**Contribution:** 3 good

**Summary:**

The authors propose a flexible representation learning approach MRL for an adaptive deployment on downstream tasks, where a single embedding can encode information at multiple granularities. An efficient variant MRL-E is also introduced with a set of common weights for nested dimensions. The coarse -to-fine representations are then applied to adaptive classification and retrieval tasks with various settings, where the method matches the performances of fixed-feature baselines. They provide as well further analysis in terms of robustness, few-shot learning and different dimensions.

**Questions:**

I am wondering if there exist any difficulties in training by introducing the nested loss.

**Limitations:**

I think the limitations mentioned in Section 6 are quite complete.

**Strengths And Weaknesses:**

1)    The paper is well-written and quite easy for readers to follow the principal ideas, though switching between appendix and main paper from time to time troubles a bit.
2)    The author provides a surprising number of experiments, covering different aspects of modalities, datasets and applications, which is quite impressive and complete.
3)    Coming to the classification task, in Tables 1,2,4, for the high dimensions, it’s interesting to see that MRL sometimes cannot achieve the same results as FF. I am wondering if some explanations could be provided.

---

> ### Author Response · Authors · 2022-07-27
> **Thank you for the positive review. Requested clarifications.**
>
> We thank the reviewer for the positive review. We are glad that the reviewer found the paper well-written, the principal ideas easy to follow, and the experiments exhaustive and impressive. Following are the clarifications requested in the review:
>
> **Switching between appendix and main paper:** Thank you for pointing this out. We will revisit the paper and its organization to minimize the back and forth. The current format of the paper tried to include all the plots in the main paper while their corresponding tables were pushed to the appendix.
>
> **Classification accuracies of the high dimensions:** The classification accuracies presented in Tables 1, 2, 4 all show that at high dimensions MRL sometimes does not have the same results as fixed feature models – MRL is at most 0.30% lower than FF models. We think this is an artifact of not having MRL-specific tuned hyperparameters. With better hyperparameters, slightly longer training, and reweighting of the MRL losses (as shown in Table 27), we can ensure that the highest dimensions of MRL models match the FF model accuracies. However, it should be noted that optimizing MRL is generally a harder problem than the baseline fixed feature models which could be a key reason for the slight dip in accuracy compared to FF at the highest dimensions.
>
> **Training difficulty of MRL:** We did not encounter any difficulties in training by introducing the nested loss for all the experiments presented in the paper. Notably, all the MRL experiments in the paper use the best hyperparameter setups of the fixed feature baselines – alleviating the need for extensive hyperparameter tuning. This makes training with MRL extremely straightforward in any existing representation learning pipeline.
>
> *We hope that this rebuttal further solidifies your positive outlook on the paper and we are happy to discuss if you need any further clarifications to increase the score and facilitate acceptance of the paper.*

---

### Official Review · Reviewer_DT9D · 2022-07-05

**Rating:** 5
**Confidence:** 5
**Soundness:** 2 fair
**Presentation:** 3 good
**Contribution:** 2 fair

**Summary:**

This paper proposes the so-called matryoshka representation learning, which can learns coarse-to-fine representations by modifying the regular learning pipeline. The matryoshka representation can encode information at different granularities, which achieves the same level of performance while being of much smaller sizes. The authors show the effectiveness of the matryoshka representations for image classification and retrieval tasks. Extensive ablation studies demonstrates the robustness and the ability of learning with limited data (few shots) and long-tailed data settings.

**Questions:**

1. It is unclear to me why the proposed algorithm can deal with out-of-distribution data and long-tailed data.

**Ethics Review Area:**

["I don’t know"]

**Limitations:**

I did not see any discussion about the limitations of this work.

**Strengths And Weaknesses:**

Strengths

1. it is an interesting idea to learn representations that can encode information of different granularities.

2. The learned representations are deployed to different downstream tasks, e.g., classification and retrieval.

3. The experiments and ablation studies are thorough and comprehensive.

Weakness

1. The idea is simple and straightforward. The major weakness of this work is the technical novelty, which makes the contribution a bit weak.

2. The proposed algorithm does not show performance advantages when the representation size is large although it performs much better when size is very small. It seems the proposed algorithm also relies on a large representations to produce SOTA performance.

---

> ### Author Response · Authors · 2022-07-28
> **Clarifications addressing the concerns and questions raised in the review.**
>
> We thank the reviewer for their time and feedback. We are glad that the reviewer found the idea interesting and the experiments to be thorough and comprehensive. Below, we answer the questions raised in the review:
>
> **“matryoshka representation can encode information at different granularities, which achieves the same level of performance while being of much smaller sizes.”:** We want to clarify that we achieve “same” performance up to minor drops (like .1% in mAP@10 for adaptive retrieval) using *adaptive* classification or retrieval, which means that for most of points we can get away with small MRL embeddings but for some points we might need to have larger embedding sizes. While it is true that smaller representation sizes (64 vs 2048) have similar accuracy (within 1%), the adaptive setups presented in the paper help in reducing the expected dimensionality even further for the same level of accuracy. This is an important distinction to note as we do not exactly match the accuracy of the highest dimensions by just using much lower dimensions – but is achievable when combined with adaptive methods.
>
> **Simplicity and Novelty:** We are glad that the reviewer found our idea to be simple and straightforward. We are unsure about the comment on technical novelty being weak. Through extensive literature review, we place MRL precisely in context of existing works and show that - to the best of our knowledge - MRL formulation and application is novel. We do agree that the idea is simple, as highlighted by the reviewer. But we view that as a significant strength of this work because the approach is generalizable to any standard architecture, domain, and loss function. Through comprehensive experiments, we do show the benefits of the approach in context of classification, retrieval, as well as for multiple domains like images, text, images-text at scale. We are happy to have a conversation on the technical novelty of the approach during the discussion phase.
>
> **Performance advantages at large representation sizes and SOTA performance:**  We do not fully understand the reviewer’s concern here.  There are two aspects to this question: a) We perform better than baselines for small dimensions. This is true because the lower dimensions get more gradient information due to the multiple losses attending to them compared to larger dimensions in MRL, b) We can achieve “nearly” the SOTA performance with much smaller dimensionality or at much smaller cost. That is, for ImageNet classification, we can get within 0.3% of SOTA with just 256 dimensional representations compared to 2048 dimensions required by the fixed feature baseline. We can in fact further reduce the expected dimensionality to 48 to achieve similar numbers by using adaptive classification. Similarly, for  retrieval, we can get within 0.1% of the SOTA mAP@10 by using MRL+adaptive retrieval.
>
> The adaptive methods slowly move up the granularities and eventually use the large representation size to achieve SOTA performance. However, the data points that require the highest dimension for any of the downstream tasks are extremely few. This makes the whole system at least an order of magnitude more efficient for deployment all while having the same performance as the strongest baseline.
> We are happy to engage in a discussion and understand your concerns in this aspect to address them clearly and make the manuscript more accessible to a general audience.
>
> **Out-of-distribution and Long-tailed data:** This is an interesting question and honestly we don’t have a very good explanation as well and believe that this would be an interesting direction to explore for future work. We thought about it extensively while analyzing the OOD and FLUID results. We hypothesize that inducing coarse-to-fine grained structure (MRL) into representation learning results in tighter packing of information in the initial dimensions – which is a result of reinforced gradient feedback from higher dimensions to lower dimensions multiple times due the MRL loss. This leaves the higher dimensions to have more capacity and handle specific (rare) semantic information that is not already taken into account. This is often lost in fixed feature models due to the diffusion of information across dimensions which tends to mix rare semantics in a hard-to-leverage fashion. However, we agree that this aspect of MRL is an interesting future work.
>
> **Limitations:** Please refer to Sections 5.1 & 6 (Lines 343 - 358), where we have noted the limitations of MRL found through ablations and provided a discussion around it along with potential future directions.
>
> *We hope that the rebuttal clarifies questions raised by the reviewer. We would be very happy to discuss any further questions about the work, and would really appreciate an appropriate increase in score if reviewers’ concerns are adequately addressed to facilitate acceptance of the paper.*

---

> > ### Comment · Reviewer_DT9D · 2022-08-05
> > **Post rebuttal**
> >
> > I would like to thank authors for their response, but I can not recommend for acceptance due the following concerns.
> >
> > 1. The limited technical novelty is a major issue which makes it below the acceptance bar of NeurIPS. I agree the idea of nested representation is interesting but still I do not anything technically novel. I do not see the application is novel either as claimed by the authors.
> >
> > 2. The contribution of this work is merely empirical, lack of insights, for instance, the performance on OOD and long-tailed data can not be explained clearly. This again makes it not acceptable for a machine learning venue.
> >
> > 3. The extensive experiments on different downstream tasks are appreciated, which unfortunately does not add any value to the technical novelty.
> >
> > 4. Though being more efficient due to the lower dimensionality of the learned representations, while the performance does not always match the original representation. It seems to me the performance of the learned representation is actually bounded by the original representation. This make me doubt about if the proposed representation really makes sense or not. At least this is not clear.

---

> > > ### Author Response · Authors · 2022-08-08
> > > **Thanks for the comments. Further clarifications.**
> > >
> > > We thank the reviewer for their comments post our rebuttal. Following are some clarifications to their comments:
> > >
> > > **Technical Novelty:** We would like to stress that our work proposes the first nested representation learning framework that is scalable and general enough to be applied to a variety of applications. The key point of our paper is extremely simple, we can design a simple multi-headed loss function for matryoshka representation learning, that performs really well in practice which we establish via extensive experiments. For example, we show speed-ups of the order of 14x without significant loss in mAP for large-scale retrieval.
> > >
> > > It would be great if the reviewer can clearly specify what they mean by "*lack of technical novelty*". To the best of our knowledge, no other work discusses such nested representations for significant impact on retrieval and classification metrics, and we have put our work in the context of the relevant literature. This would also help us in improving the paper further.
> > >
> > > **Insights:** We indeed agree that our paper establishes the utility of matryoshka representations empirically via extensive experiments across multiple applications on standard benchmarks. We also agree that more theoretical/grounded insights would be quite useful. But we would like to highlight that, in this area, generally theoretical insights are hard to derive, and a lot of progress has been made by more empirically established techniques (e.g., deep learning itself doesn't have a very strong theoretical basis or insights for a lot of the observations, similarly HNSW methods used extensively in retrieval still do not have compelling theoretical basis).
> > >
> > > We strongly believe that empirical results and observations in this paper form a basis for interesting future work on explaining the insights and implications. Section 5 on further analysis and ablations in the paper does provide initial insights and some understanding of the proposed representations. Akin to a lot of empirical ML papers published in NeurIPS, MRL provides a simple algorithm with an extensive experimental evaluation which makes an extremely strong case for real-world usage.
> > >
> > > **Matching performance of higher dimension with lower dimensional representation:** We agree that our method's accuracy can at most match the original representation. But we would like to highlight that even at almost 20x lower-dimensional representations, we are able to get within 0.5% of accuracy of the original representation. Furthermore, this translates to 14x reduction in wall-clock time for retrieval of relevant items which is a significant gain and is critical for practical applications all while not losing anything on mAP@10 metric. Finally, this also allows us to train for larger representation sizes without worrying about deployment constraints, as we can leverage the improved accuracy through much more efficient (expected) lower dimensional representations which will make real-world deployment feasible.
> > >
> > > We are still not sure as to what is the confusion here. The proposed representation matches the "upper bound" original representation accuracy at a much cheaper cost (*more efficient*). Improving the efficiency of retrieval that uses deep representations has been an important research direction with several papers being published in NeurIPS (Chen et al., NeurIPS 2021, Subramanya et al., NeurIPS 2019 etc.,) -- note that these methods are complementary to ours. We think bringing efficiency alongside similar accuracy of an existing method is a significant contribution to real-world deployment. Please do let us know if the proposed representation brings value to the table and makes sense.
> > >
> > > *We are happy to have further discussion in the next couple of days to clarify any other concerns and improve the manuscript.*

---

> > ### Comment · Reviewer_DT9D · 2022-08-08
> > **Final**
> >
> > Thank you for your further clarification. I will upgrade my score.

---

> > > ### Author Response · Authors · 2022-08-08
> > > **Thank you.**
> > >
> > > We thank the reviewer for upgrading the score.

---

### Official Review · Reviewer_HvJ7 · 2022-07-07

**Rating:** 6
**Confidence:** 3
**Soundness:** 3 good
**Presentation:** 4 excellent
**Contribution:** 3 good

**Summary:**

The paper introduces "Matryoshka Representations" which are a way to encode dimensions over multiple scales such that for a chosen embedding dimension d, there are log(d) embeddings of increasingly smaller size that can encode the same information as the original d-dimensional vector, with minimal loss in performance of the downstream task. The paper thoroughly validates MRs on an impressively large variety of tasks spanning over challenging and large scale datasets, and different modalities (vision+language, vision, language).

**Questions:**

Please see strengths/ weaknesses.

**Limitations:**

The authors do not explicitly discuss societal impacts of their work, but its unclear if the work has direct impacts. It would be useful for the community if the authors commented/showed if the lower dimensional representations are more or less susceptible of encoding biases in their representations, as its important to discuss such details. For practical purposes, is there a tradeoff between the dimensionality of the representations and the tendency to encode biases.

**Strengths And Weaknesses:**

Strengths:
- the main strength of the paper is that it proposes a simple and practical solution for the problem, which can be used in many downstream tasks
- The paper is written very clearly.
- 14 x faster retrieval is a very significant improvement
- The experiments are exhaustive and satisfying, for an approach that offers a lot of generality.

Weaknesses:
- Outside of increasing the speed for retrieval tasks, I am not sure what other practical applications of MRs are there.
- I would like to see the authors discuss the research done on anytime neural networks[1]. They are not similar in methodology, but try to solve a very similar problem.
- The improvement over FF is impressive, albeit unclear to me, why there is still improvements are higher dimensions (~64+). I would think/hope that at around 64 dim, the amount of information starts to become noisy. I would also
- I would be very interested in these two baselines on any task:
    - VIB [2] reformulated for this task, to see if its possible that there are any direct or indirect connections if the low-dim representations learned using VIB can be useful.
    - Taking the full d-dim embedding and projecting it to a lower dim space using a random projection and seeing if that works. According to Gram-Schmidt, the low-dim representations should be usefully informative, and it was tried in [3] to some success. It can be a useful baseline for the paper as well.


[1] https://arxiv.org/abs/1708.06832
[2] https://arxiv.org/abs/1612.00410
[3] https://arxiv.org/abs/1910.13540

---

> ### Author Response · Authors · 2022-07-28
> **Thank you for the thorough review. Requested Clarifications -- Part 2/2**
>
> Continuation of [Part 1/2](https://openreview.net/forum?id=9njZa1fm35&noteId=XbaARUESk8u).
>
> **Random Projections:** We think you mean Johnson–Lindenstrauss (JL) when you mention Gram-Schmidt, as JL lemma suggests random projections to be a great way to reduce dimensionality while holding onto the similarity information. We did run random projections as a baseline but did not include them in the paper due to their abysmal performance compared to random feature selection and SVD – which is often the case for large-scale semantic search. However, please find the results of gaussian random projection with 1-NN accuracy (%) compared to SVD on FF - 2048 model in the table below and we will include random projection as a baseline in the next revision of the paper. We think SVD is a much better projection technique than random for dimensionality reduction.
>
> | Representation Size      | Random Projection | SVD |
> | :-----------: | :-----------: | :-----------: |
> | 8      |    0.088    | 19.14|
> | 16   |    0.093    | 46.02|
> | 32      |    0.12    | 60.78|
> | 64   |    0.10     | 67.04|
> | 128      |    0.14    | 69.63|
> | 256   |    0.07     | 70.67|
> | 512   |     0.11    | 71.06|
>
> **Societal Impact:** This was an oversight on our end and completely missed explaining the potential social impact in the checklist. The reviewer is correct in stating that our work does not have any additional negative societal impact on top of what representation learning already has. We will incorporate the societal impact into the next revision of the manuscript.
>
> We agree that the study on the tradeoff between representation size and the tendency to encode biases is an interesting direction. Along the lines of Hooker et al 2019, 2020 (https://arxiv.org/abs/1911.05248, https://arxiv.org/abs/2010.03058), we did some experiments to see the trends between the classes and their variance in accuracy with the change of dimensionality. A part of this was also presented as part of the “Disagreement across Dimensions” (Lines 334 - 342) which discusses superclass accuracy and hard instances. We will also include the per-class accuracy trends (some lose more than others) with dimensionality in the next revision of the paper.
>
> All these tie well into the aspect of encoding biases and investigating them with the lens of information bottleneck capacities is an exciting future direction.
>
> *We hope that this rebuttal further solidifies your positive outlook on the paper and we are happy to discuss if you need any further clarifications to increase the score and facilitate acceptance of the paper.*

---

> > ### Comment · Reviewer_HvJ7 · 2022-08-05
> > **Thanks for the detailed and thorough responses**
> >
> > Yes, I did mean JL - thanks for the correction. Indeed surprised to see the abysmal results with JL, but I would add this experiment in the appendix for the final copy.
> >
> > Most of my concerns have been adequately satisfied. Although I have some concerns about the large scale impact especially in terms of application, since speed and latency issues are often solved by pruning the search space, instead of decreasing the dimensionality of the representation, I still think there is very solid insight in this work. Updating the score.

---

> > > ### Author Response · Authors · 2022-08-05
> > > **Thanks again!**
> > >
> > > Dear reviewer,
> > >
> > > We are glad to hear that most of your concerns have been addressed. We will add the random projection experiment in the appendix for the camera-ready.
> > >
> > > **Large-scale Impact:** Thanks for finding our work insightful. We agree that at web-scale (billion-trillion data points) often the speed and latency issues are solved by efficient pruning of search space through Approximate Nearest Neighbors Search (ANNS). However, despite that that there is still a linear dependence on the dimensionality of the representation (*d*) which MRL handles extremely well in expectation through adaptivity.
> > >
> > > Another key thing to note is that ANNS does search space pruning on a compressed index that is built post-hoc and not learned. The lower dimensions of MRL also tackle this sub-optimality by providing an accurate, learned low-d index that is much easier to do search space pruning on. Finally, all of the techniques proposed in the paper are complementary to the current SOTA methods for search space pruning and ANNS -- while further improving efficiency with almost no accuracy drop. This was demonstrated in the real-world implementation of adaptive retrieval using ANNS for the initial shortlisting using lower dimensions (eg., 8 and 16) followed by exact reranking using the more informative dimensions. We show that this results in no accuracy drop whatsoever compared to an exact L2 search (brute-force k-NN).
> > >
> > >
> > > *Hope this further clarifies the last concern brought up by the reviewer, we thank them again for their time and constructive feedback*.

---

> ### Author Response · Authors · 2022-07-28
> **Thank you for the thorough review. Requested Clarifications -- Part 1/2**
>
> We thank the reviewer for the detailed review. We are glad that the reviewer found the proposed solution simple and practical while being general, the paper to be well written, the experiments to be exhaustive and satisfying and the real-world improvements in retrieval to be significant. Following are the clarifications requested in the review:
>
> **Other Practical Applications of MRs:** We think increasing speed for retrieval tasks is a significant application, as you also agree in the strengths. MRs allow training/maintaining/inferring with *one* network and still allow for flexibility for downstream tasks powered through classification and retrieval. This is very critical in practice, because compute environments, latency requirements, and accuracy constraints for downstream tasks vary significantly. So MRL can allow maintaining one representation to serve multiple downstream tasks satisfactorily. In addition, MRs can be used for adaptive classification, quantifying the hardness of instances, and improved few-shot and long-tail performance all while maintaining the same robustness as the fixed feature baselines.
>
> Additionally, MRs can also be used for adaptive latency-based transmission over computer networks. This would involve sending compressed representations of the data to be reconstructed depending on network bandwidth – MRs naturally solve this problem with the coarse-to-fine grained flexibility baked into them.
> We are happy to discuss more practical applications of MRs and are looking forward to your feedback.
>
> **Anytime Neural Networks (ANNs):** As you mentioned, we are solving similar problems from different angles. ANNs aims at reducing the total inference cost by adaptive exiting at various depths. However, in the case of large-scale search and retrieval, this would involve storing multiple representations across depths in the database – which is prohibitively expensive for search and storage. But MRL is agnostic to any representation learning and can be combined with ANNs across all depths.
>
> However, the more interesting thing from the paper (https://arxiv.org/abs/1708.06832) is the aspect of the adaptive weight scheme for train losses – which is directly applicable to MRL’s own loss weightings. We thank the reviewer for this pointer (will cite in the next revision) and this would be very useful in future work involving the right weightings of loss functions in MRL and multi-objective MRL.
>
> **Performance improvement at higher dimensions:** We agree that improvement over FF is an interesting observation. We hypothesize it is due to multi-fold gradients of information that pass through the lower dimensions as they are shared across higher dimensions. For example, even 128 dims would get gradient feedback when we are also minimizing the losses for 256, 512, 1024 and 2048 dim. At the same time, accuracy is not saturated yet, resulting in slightly higher accuracy than the FF models. We find this to be interesting as well and are willing to have a longer discussion in this regard.
>
> **Variational Information Bottleneck (VIB):** VIB is a really interesting line of work – thanks for pointing us to it. It is a valid setup to have VIB adapted to a general representation learning setup and see if the bottleneck size is *much* lower than a vanilla neural network. Viewing VIB as a regularization, it can be easily combined with MRL making both the techniques complementary if need be. Lastly, training variational methods at large-scale are more expensive compared to usual training recipes; however, we think that VIB + MRL is an extremely interesting direction for empirically analyzing information bottlenecks.
>
> **Rest of the clarifications in the comment below** -- [Part 2/2](https://openreview.net/forum?id=9njZa1fm35&noteId=uUmACsunpG7)

---

### Official Review · Reviewer_pjGy · 2022-07-12

**Rating:** 7
**Confidence:** 3
**Soundness:** 3 good
**Presentation:** 3 good
**Contribution:** 3 good

**Summary:**

In this work, the authors propose Matryoshka Representations, a flexible coarse-to-fine representation that can be applied to multiple downstream tasks based on the computational budget. The authors demonstrate that utilizing the proposed Matryoshka Representations leads to a substantial reduction in embedding size for large-scale retrieval tasks. Finally, the authors have shown that the proposed representations can be incorporated into web-scale dataset experiments.

**Questions:**

- Have the authors experimented with arbitrary granularities? For instance, m random numbers can be generated up to the highest embedding size and these random numbers can be used for nesting dimensions. It will be interesting to see how the embeddings interpolate if the features are learned using such an arbitrary granularity.

- Have the authors tried lower than 8 dimensions?

- Any insight into why tuning the relative importance weight of dimension 16 did not lead to improvement of dimension 32 whereas only increasing the weight of dimension 8 led to a noticeable improvement of dimension 16.

**Limitations:**

The limitations of the proposed work have been adequately addressed. However, no discussion about the potential negative soceital impact has been provided in the main text and supplementary materials.

**Strengths And Weaknesses:**

Strengths:

- The proposed solution is simple and intuitive.
- The learned low-dimensional representations are as good as the models trained independently with that particular dimension without any additional computational overhead.
- Significantly improves the efficiency of large-scale classification and retrieval tasks.
- Allows for adaptive classification with variable embedding size.
- Achieves promising improvement on long-tail continual learning experiments.
- Matryoshka Representations can be easily extended to different modalities and even to web-scale datasets.

Weaknesses:
- No major weakness. The idea is simple and the results are compelling.

---

> ### Author Response · Authors · 2022-07-29
> **Thank you for the positive and thorough review. Requested experimental clarifications. Part 2/2**
>
> Continuation of [Part 1/2](https://openreview.net/forum?id=9njZa1fm35&noteId=JFeMFthioB).
>
> **Performance improvement on next granularities based on tuning relative importance:**  This is an interesting observation and we do not think we have a concrete insight into why this is happening. Our hypothesis is that this is highly dependent on hyperparameters and given that we use the same setting for every experiment (even the relative importance ablations), we might not be looking at the best possible outcome. We think that with the right hyperparameters and better training routines, the accuracy of the next granularity will benefit significantly from improving the lower representations.
>
> **Societal Impact:** This was an oversight on our end and completely missed explaining the potential social impact in the checklist. Our work does not have any additional negative societal impact on top of what representation learning already has. We will incorporate the societal impact into the next revision of the manuscript. We would also ask the reviewer to check the response on societal impact in [our rebuttal to reviewer HvJ7](https://openreview.net/forum?id=9njZa1fm35&noteId=uUmACsunpG7).
>
> *We hope that this rebuttal further solidifies your positive outlook on the paper and we are happy to discuss if you need any further clarifications to increase the score and facilitate acceptance of the paper.*

---

> > ### Comment · Reviewer_pjGy · 2022-08-08
> > **Post Rebuttal Response**
> >
> > I have read the rebuttal and the reviews from other reviewers. I really appreciate the explanations and additional experimental results provided in the rebuttal. The authors' response has adequately addressed all of my concerns. Just to reiterate my initial review, I do believe the empirical results and insights provided in this work are significant and will be of interest to the community. Therefore, I will be keeping my initial score.

---

> > > ### Author Response · Authors · 2022-08-08
> > > **Appreciate the support, thank you again!**
> > >
> > > Dear reviewer,
> > >
> > > We are grateful for your kind words and support. We are glad that the rebuttal and additional experiments adequately addressed your concerns. Finally, thanks for recognizing the value of our work -- ideas, empirical results, and insights -- and its potential impact both in research and real-world deployment.
> > >
> > > Thank you,
> > >
> > > Authors.

---

> ### Author Response · Authors · 2022-07-29
> **Thank you for the positive and thorough review. Requested experimental clarifications. Part 1/2**
>
> We thank the reviewer for the detailed and positive review. We are glad that the reviewer found the proposed solution simple and intuitive while being task and scale agnostic, the improved efficiency significant, adaptive methods interesting and improvement in long-tail continual learning promising. Following are the clarifications requested in the review:
>
> **Arbitrary granularities:** Thank you for the suggestion on arbitrary granularities. There 2 reasons for choosing logarithmic granularities over arbitrary granularities: a) The accuracy improvement vs representation size is not linear but is more logarithmic (as shown by the FF models) – This means, that optimizing for granularities increasing in a non-logarithmic fashion would be sub-optimal both for maximum performance and expected efficiency. b) If we have “m'' arbitrary granularities, the expected cost of the linear classifier to train MRL would scale O(L*(m^2)) while logarithmic granularities result in O(L*2log(d)) space and compute costs. These two reasons put together make logarithmic granularities a strong empirically driven design choice.
>
> However, we have run an experiment with uniform (*MRL-Uniform*) interval granularities [8, 212, 416, 620, 824, 1028, 1232, 1436, 1640, 1844, 2048] as shown in the table below. We have evaluated the interpolation at the usual (*MRL-log*) dimensions [16, 32, 64, 128, 256, 512, 1024, 2048] for ease of comparison to reported numbers using 1-NN accuracy (%).
>
> We observe that the interpolation does happen, but is not strong as logarithmic granularities. The biggest difference is at the lowest dimensions as logarithmic spacing helps in tighter packing of information in the initial dimensions increasing the accuracy significantly. It is also evident that having granularities at higher dimensions does not help in significant accuracy improvement -- due to accuracy saturation which is often logarithmic w.r.t representation size as shown by FF models. Note that the slight improvement at higher dimensions for uniform beyond 512 is due to the multiple granularities around them compared to just 3 for the logarithmic -- which are not useful in practice for efficiency.
>
> | Representation Size | MRL-Uniform | MRL-log |
> |:-------------------:|:-----------:|:-------:|
> |          8          |      58.44       |  62.19  |
> |       16 (int)      |      61.11       |  67.91  |
> |       32 (int)      |      63.82      |  69.46  |
> |       64 (int)      |      66.44       |  70.17  |
> |      128 (int)      |      68.71       |  70.52  |
> |         212         |       69.95      |    --   |
> |      256 (int)      |     70.06        |  70.62  |
> |         416         |       70.84      |    --   |
> |      512 (int)      |      70.98       |  70.82  |
> |         620         |       71.16      |    --   |
> |         824         |        71.30     |    --   |
> |      1024 (int)     |     71.37        |  70.89  |
> |         1028        |      71.39       |    --   |
> |         1232        |       71.48      |    --   |
> |         1436        |       71.49      |    --   |
> |         1640        |      71.50       |    --   |
> |         1844        |       71.48      |    --   |
> |         2048        |     71.44        |  70.97  |
>
> **Lower than 8 dimensions:** Yes, we have tried working with lower than 8-dimensions. However, for ImageNet scale and beyond, we found that using lower than 8-dimensions in MRL does not affect the accuracies of other granularities significantly.  However, lower than 8-dimension granularity have low accuracies which often can not be used during deployment meaningfully due to the hardness in the training of the lowest dimension. We also observe a small dip in accuracy at higher dimensions which we attribute to the joint loss that now also includes the harder optimization of lower than 8-dimensions. Lastly, we think the dimensionality of 8 is an empirically validated design choice due to the considerable accuracy it provides along with the ease of training.
>
> Here are the accuracy numbers for an experiment where we used MRL between 4-2048 (*MRL-4*) and 6-2048 (*MRL-8*) dimensions on ImageNet using ResNet50 compared to reported numbers (*MRL-8*) using linear classification (Top-1 accuracy(%)).
>
> | Representation Size | MRL-4 | MRL-6 | MRL-8 |
> |:-------------------:|:-----:|:-----:|-------|
> |          4          | 27.25 | --    | --    |
> |          6          | --    | 58.71 | --    |
> |          8          | 66.86 | 67.55 | 66.63 |
> |          16         | 73.36 |  73.10 | 73.53 |
> |          32         | 74.82 | 74.49 | 75.03 |
> |          64         | 75.51 | 75.32 | 75.82 |
> |         128         | 75.93 | 75.61 |  76.30 |
> |         256         | 76.08 | 75.82 | 76.47 |
> |         512         | 76.31 | 75.93 | 76.65 |
> |         1024        | 76.38 | 76.04 | 76.76 |
> |         2048        | 76.43 | 76.12 |  76.80 |
>
> Rest of the clarifications in the comment below -- [Part 2/2](https://openreview.net/forum?id=9njZa1fm35&noteId=Pm7UdhvIjGv)

---

### Meta-Review · Area_Chair_zaaP · 2022-09-04

**Recommendation:** Accept
**Confidence:** Certain

**Metareview:**

This paper proposes a Matryoshka Representation Learning paradigm to learn representations at multiple granularities, which can adapt to downstream tasks with different computational budgets. All the reviewers find the idea simple and interesting, and acknowledge that the experiments are thorough and impressive. The authors were successful at addressing the reviewers' concerns. Overall, the meta-reviewer recommends acceptance of the paper.

**Award:**

No

---

### Decision · Program_Chairs · 2022-09-14

Accept